# ON THE CONVERGENCE OF THE MONTE CARLO EXPLORING STARTS ALGORITHM FOR REINFORCEMENT LEARNING

**Che Wang**[1,2]    **Shuhan Yuan**[1]    **Kai Shao**[1]    **Keith Ross**[1*]

[1] New York University Shanghai
[2] New York University

### ABSTRACT

A simple and natural algorithm for reinforcement learning (RL) is Monte Carlo Exploring Starts (MCES), where the Q-function is estimated by averaging the Monte Carlo returns, and the policy is improved by choosing actions that maximize the current estimate of the Q-function. Exploration is performed by "exploring starts", that is, each episode begins with a randomly chosen state and action, and then follows the current policy to the terminal state. In the classic book on RL by Sutton & Barto (2018), it is stated that establishing convergence for the MCES algorithm is one of the most important remaining open theoretical problems in RL. However, the convergence question for MCES turns out to be quite nuanced. Bertsekas & Tsitsiklis (1996) provide a counter-example showing that the MCES algorithm does not necessarily converge. Tsitsiklis (2002) further shows that if the original MCES algorithm is *modified* so that the Q-function estimates are updated at the same rate for all state-action pairs, and the discount factor is strictly less than one, then the MCES algorithm converges. In this paper we make headway with the original and more efficient MCES algorithm given in Sutton & Barto (1998), establishing almost sure convergence for Optimal Policy Feed-Forward MDPs, which are MDPs whose states are not revisited within any episode when using an optimal policy. Such MDPs include a large class of environments such as all deterministic environments and all episodic environments with a timestep or any monotonically changing values as part of the state. Different from the previous proofs using stochastic approximations, we introduce a novel inductive approach, which is very simple and only makes use of the strong law of large numbers.

## 1 INTRODUCTION

Perhaps the most famous algorithm in tabular reinforcement learning is the so-called Q-learning algorithm. Under very general conditions, it is well known that the Q-learning converges to the optimal Q-function with probability one (Tsitsiklis, 1994; Jaakkola et al., 1994). Importantly, in order to guarantee convergence for Q-learning, it is only required that every state-action pair be visited infinitely often. Furthermore, as discussed in the related work, Q-learning converges for the infinite-horizon discounted problem as well as for the non-discounted terminal-state problem (also known as the stochastic shortest-path problem).

The Q-learning algorithm is inspired by dynamic programming and uses back-ups to update the estimates of the optimal Q-function. An alternative methodological approach, which does not use back-ups, is to use the Monte Carlo episodic returns to estimate the values of the Q-function. In order for such an algorithm to succeed at finding an optimal policy, the algorithm must include some form of exploration. A simple form of exploration is "exploring starts," where at the beginning of each episode, a random state-action pair is chosen. In the classic book on reinforcement learning by Sutton & Barto (2018), the authors describe such an algorithm, namely, Monte Carlo Exploring

---

*Correspondence to: Keith Ross <keithwross@nyu.edu>.

Starts (MCES). In MCES, after a (random-length) episode, the Q-function estimate is updated with the Monte Carlo return for each state-action pair along the episode, and the policy is improved in the usual fashion by setting it to the argmax of the current Q-function estimate. Exploration is performed by exploring starts, where the initial state-action pairs may be chosen with any distribution.

We briefly remark here that AlphaZero is a Monte Carlo algorithm in that it runs episodes to completion and uses the returns from those episodes for the targets in the loss function (Silver et al., 2018). AlphaZero additionally uses function approximators and planning (Monte Carlo Tree Search), and is thus much more complicated than MCES. But AlphaZero is nonetheless fundamentally a Monte Carlo algorithm rather than a Q-learning-based algorithm. We mention AlphaZero here in order to emphasize that Monte Carlo algorithms are indeed used in practice, and it is therefore important to gain a deep understanding of their underlying theoretical properties. Additional discussion is provided in Appendix C.

Since Q-learning converges under very general conditions, a natural question is: does MCES converge under equally general conditions? In the 1996 book, Bertsekas & Tsitsiklis (1996) provide a counter-example showing that the MCES algorithm does not necessarily converge. See also Liu (2020) for numerical results in this direction. Thus, we see that the MCES convergence problem is fundamentally trickier than the Q-learning convergence problem. Instead of establishing a very general result as in Q-learning, we can at best establish convergence for a broad class of special-cases.

Sutton and Barto write at the end of Section 5.3: "In our opinion, this is one of the most fundamental open theoretical questions in reinforcement learning". This paper is focused on this fundamental question. Although other questions, such as rates of convergence and regret bounds, are also important, in this paper our goal is to address the fundamental question of convergence.

Tsitsiklis (2002) made significant progress with the MCES convergence problem, showing that almost sure convergence is guaranteed if the following three conditions hold: $(i)$ the discount factor is strictly less than one; $(ii)$ the MCES algorithm is modified so that after an episode, the Q-function estimate is updated with the Monte Carlo return *only for the initial state-action pair of the episode*; and $(iii)$ the algorithm is further modified so that the initial state-action pair in an episode is chosen with a uniform distribution. As in the proof of Q-learning, Tsitsiklis's proof is based on stochastic approximations. The conditions $(ii)$ and $(iii)$ combined ensure that the Q function estimates are updated at the same average rate for all state-action pairs, and both conditions appear to be crucial for establishing convergence in the proof in Tsitsiklis (2002). However, these two conditions have the following drawbacks:

- Perhaps most importantly, condition $(ii)$ results in a substantially less efficient algorithm, since only one Q-function value is updated per episode. The original Sutton and Barto version is more efficient since after each episode, many Q-values are typically updated rather than just one. (We also note as an aside that AlphaZero will also collect and use Monte Carlo return for all states along the episode, not just for the first state in the episode, as discussed on page 2 of Silver et al. (2017), also see discussion in Appendix.)

- Similar to the idea of importance sampling, one may want to use a non-uniform distribution for the starting state-action pairs to accelerate convergence.

- In some cases, we may not have access to a simulator to generate uniform exploring starts. Instead, we may run episodes by interacting directly with the real environment. Such natural interactions may lead to starting from every state, but not uniformly. An example would be playing blackjack at a casino rather than training with a simulator.

In this paper we provide new convergence results and a new proof methodology for MCES. Unlike the result in Tsitsiklis (2002), the results reported here do not modify the original MCES algorithm and do not require any of the conditions $(i) - (iii)$. Hence, our results do not have the three drawbacks listed above, and also allow for no discounting (as in the stochastic shortest path problem). However, our proofs require restrictions on the dynamics of the underlying MDP. Specifically, we require that under the optimal policy, a state is never revisited. This class of MDPs includes stochastic feed-forward environments such as Blackjack (Sutton & Barto, 2018) and also all deterministic MDPs, such as gridworlds (Sutton & Barto, 2018), Go and Chess (when played against a fixed opponent policy), and the MuJoCo environments (Todorov et al., 2012) (Episodic MuJoCo tasks fall into the category of OPFF MDPs because the MuJoCo simulation is deterministic). More examples

are provided in appendix C. Moreover, if the trajectory horizon is instead fixed and deterministic, we show that the original MCES algorithm *always* converges (to a time-dependent) optimal policy, without any conditions on the dynamics, initial state-action distribution or the discount factor.

Importantly, we also provide a new proof methodology. Our proof is very simple, making use of only the Strong Law of Large Numbers (SLLN) and a simple inductive argument. The proof does not use stochastic approximations, contraction mappings, or martingales, and can be done in an undergraduate course in machine learning. We believe that this new proof methodology provides new insights for episodic RL problems.

In addition to the theoretical results, we present numerical experiments that show the original MCES can be much more efficient than the modified MCES, further highlighting the importance of improving our understanding on the convergence properties of the original MCES algorithm.

## 2 RELATED WORK

Some authors refer to an MDP with a finite horizon $H$ as an episodic MDP. For finite horizon MDPs, the optimal Q-function and optimal policy are in general non-stationary and depend on time. Here, following Sutton & Barto (2018), we instead reserve the term *episodic MDPs* for MDPs that terminate when the terminal state is reached, and thus the episode length is not fixed at $H$ and may have a random length. Moreover, for such terminal-state episodic MDPs, under very general conditions, the optimal Q-function and policy are stationary and do not depend on time (as in infinite-horizon discounted MDPs). When the dynamics are known and the discount factor equals 1, the episodic optimization problem considered here is equivalent to the stochastic shortest path problem (SSPP) (see Bertsekas & Tsitsiklis (1991) and references therein; also see Chapter 2 of Bertsekas (2012)). Under very general conditions, value iteration converges to the optimal value function, from which an optimal stationary policy can be constructed.

Convergence theory for RL algorithms has a long history. For the infinite-horizon discounted criterion, by showing that Q-learning is a form of stochastic approximations, Tsitsiklis (1994) and Jaakkola et al. (1994) showed that Q-learning converges almost surely to the optimal Q-function under very general conditions. There are also convergence results for Q-learning applied to episodic MDPs as defined in this paper with discount factor equal to 1. Tsitsiklis [8, Theorems 2 and 4(c)] proved that if the sequence of Q-learning iterates is bounded, then Q-learning converges to the optimal Q values almost surely. Yu & Bertsekas (2013) prove that the sequence of Q-learning iterates is bounded for episodic MDPs with or without non-negativity assumptions, fully establishing the convergence of Q-learning for terminal-state episodic RL problems.

This paper is primarily concerned with the convergence of the MCES algorithm. In the Introduction we reviewed the important work of Sutton & Barto (1998), Bertsekas & Tsitsiklis (1996), and Tsitsiklis (2002). Importantly, unlike Q-learning, the MCES algorithm is not guaranteed to converge for all types of MDPs. Indeed, in Section 5.4 of Bertsekas & Tsitsiklis (1996), Example 5.12 shows that MCES is not guaranteed to converge for a continuing task MDP. However, if the algorithm is modified, as described in the Introduction, then convergence is guaranteed (Tsitsiklis, 2002). Recently, Chen (2018) extended the convergence result in Tsitsiklis (2002) to the undiscounted case, under the assumption that all policies are proper, that is, regardless of the initial state, all policies will lead to a terminal state in finite time with probability one. More recently, Liu (2020) relaxed the all policies being proper condition. As in Tsitsiklis (2002), both Chen (2018) and Liu (2020) assume conditions $(ii) - (iii)$ stated in the introduction, and their proofs employ the stochastic approximations methodology in Tsitsiklis (1994). The results we develop here are complementary to the results in Tsitsiklis (2002), Chen (2018), and Liu (2020), in that they do not require the strong algorithmic assumptions $(ii) - (iii)$ described in the Introduction, and they use an entirely different proof methodology.

In this work we focus on the question of convergence of the MCES problem. We briefly mention, there is also a large body of (mostly orthogonal) work on rates of convergence and regret analysis for Q-learning (e.g. see Jin et al. (2018)) and also for Monte Carlo approaches (e.g., see Kocsis & Szepesvári (2006) Azar et al. (2017)). To the best of our knowledge, these regret bounds assume finite-horizon MDPs (for which the optimal policy is time-dependent) rather than the terminal-state episodic MDPs considered here.

## 3 MDP FORMULATION

Following the notation of Sutton & Barto (2018), a finite Markov decision process is defined by a finite state space $\mathcal{S}$ and a finite action space $\mathcal{A}$, reward function $r(s, a)$ mapping $\mathcal{S} \times \mathcal{A}$ to the reals, and a dynamics function $p(\cdot|s, a)$, which for every $s \in \mathcal{S}$ and $a \in \mathcal{A}$ gives a probability distribution over the state space $\mathcal{S}$.

A (deterministic and stationary) **policy** $\pi$ is a mapping from the state space $\mathcal{S}$ to the action space $\mathcal{A}$. We denote $\pi(s)$ for the action selected under policy $\pi$ when in state $s$. Denote $S_t^\pi$ for the state at time $t$ under policy $\pi$. Given any policy $\pi$, the state evolution becomes a well-defined Markov chain with transition probabilities

$$P(S_{t+1}^\pi = s'|S_t^\pi = s) = p(s'|s, \pi(s))$$

### 3.1 RETURN FOR RANDOM-LENGTH EPISODES

As indicated in Chapters 4 and 5 of Sutton & Barto (2018), for RL algorithms based on MC methods, we assume the task is episodic, that is "experience is divided into episodes, and all episodes eventually terminate no matter what actions are selected." Examples of episodic tasks include "plays of a game, trips through a maze, or any sort of repeated interaction". Chapter 4 of Sutton & Barto (2018) further states: "Each episode ends in a special state called the *terminal state*, followed by a reset to a standard starting state or to a sample from a standard distribution of starting states".

The "Cliff Walking" example in Sutton & Barto (2018) is an example of an "episodic MDP". Here the terminal state is the union of the goal state and the cliff state. Although the terminal state will not be reached by all policies due to possible cycling, it will clearly be reached by the optimal policy. Another example of an episodic MDP, discussed at length in Sutton and Barto, is "Blackjack". Here we can create a terminal state which is entered whenever the player sticks or goes bust. For Blackjack, the terminal state will be reached by all policies. Let $\tilde{s}$ denote the terminal state. (If there are multiple terminal states, without loss in generality they can be lumped into one state.)

When using policy $\pi$ to generate an episode, let

$$T^\pi = \min\{ t \ : \ S_t^\pi = \tilde{s}\} \tag{1}$$

be the time when the episode ends. The expected total reward when starting in state $s$ is

$$v_\pi(s) = E[\sum_{t=0}^{T^\pi} r(S_t^\pi, \pi(S_t^\pi))|S_0^\pi = s] \tag{2}$$

Maximizing (2) corresponds to the stochastic shortest path problem (as defined in Bertsekas & Tsitsiklis (1991); Bertsekas (2012)), for which there exists an optimal policy that is both stationary and deterministic (for example, see Proposition 2.2 of Bertsekas (2012)).

At the end of this paper we will also consider the finite horizon problem of maximizing:

$$v_\pi^H(s) = E[\sum_{t=0}^{H} r(S_t^\pi, \pi(S_t^\pi, t))|S_0^\pi = s] \tag{3}$$

where $H$ is a fixed and given horizon. For this criterion, it is well-known that optimal policy $\pi(s) = (\pi(s, 1), \ldots, \pi(s, H))$ is non-stationary. For the finite-horizon problem, it is not required that the MDP have a terminal state.

### 3.2 CLASSES OF MDPS

We will prove convergence results for important classes of MDPs. For any MDP, define the MDP graph as follows: the nodes of the graph are the MDP states; there is a directed link from state $s$ to $s'$ if there exists an action $a$ such that $p(s'|s, a) > 0$.

We say an environment is **Stochastic Feed-Forward (SFF)** if a state cannot be revisited within any episode. More precisely, the MDP is SFF if its MDP graph has no cycles. Note that transitions are permitted to be stochastic. SFF environments occur naturally in practice. For example, the

Blackjack environment, studied in detail in Sutton & Barto (2018), is SFF. We say an environment is **Optimal Policy Feed-Forward (OPFF)** if under any optimal policy a state is never re-visited. More precisely, construct a sub-graph of the MDP graph as follows: each state $s$ is a node in the graph, and there is a directed edge from node $s$ to $s'$ if $p(s'|s, a^*) > 0$ for some optimal action $a^*$. The MDP is OPFF if this sub-graph is acyclic.

An environment is said to be a **deterministic environment** if for any state $s$ and chosen action $a$, the reward and subsequent state $s'$ are given by two (unknown) deterministic functions $r = r(s, a)$ and $s' = g(s, a)$. Many natural environments are deterministic. For example, in Sutton & Barto (2018), environments Tic-Tac-Toe, Gridworld, Golf, Windy Gridworld, and Cliff Walking are all deterministic. Moreover, many natural environments with continuous state and action spaces are deterministic, such as the MuJoCo robotic locomotion environments (Todorov et al., 2012). It is easily seen that all SFF MDPs are OPFF, and all deterministic MDPs for which the optimal policy terminates w.p.1 are OPFF.

## 4 Monte Carlo with Exploring Starts

The MCES algorithm is given in Algorithm 1. The MCES algorithm is a very natural and simple algorithm, which uses only Monte Carlo returns and no backups for updating the Q-function and training the policy. A natural question is: does it converge to the optimal policy? As mentioned in the Introduction, unlike for Q-learning, it does not converge for general MDPs. We instead establish convergence for important classes of MDPs.

Algorithm 1 is consistent with the MCES algorithm in Sutton & Barto (2018) and has the following important features:

- $Q(S_t, A_t)$ is updated for every $S_t, A_t$ pair along the episode, and not just for $S_0, A_0$ as required in (Tsitsiklis, 2002).

- The initial state and action can be chosen arbitrarily (but infinitely often), and does not have to be chosen according to a uniform distribution as required in (Tsitsiklis, 2002).

- For simplicity, we present the algorithm with no discounting (i.e., $\gamma = 1$). However, the subsequent proofs go through for any discount factor $\gamma \leq 1$. (The proof in Tsitsiklis (2002) requires $\gamma < 1$.)

We emphasize that although the results in Tsitsiklis (2002) require restrictive assumptions on the algorithm, the results in Tsitsiklis (2002) hold for general MDPs. Our results do not make restrictive algorithmic assumptions, but only hold for a sub-class of MDPs.

---

**Algorithm 1** MCES

1: Initialize: $\pi(s) \in \mathcal{A}, Q(s, a) \in \mathbb{R}$, for all $s \in \mathcal{S}, a \in \mathcal{A}$, arbitrarily;
   $Returns(s, a) \leftarrow$ empty list, for all $s \in \mathcal{S}, a \in \mathcal{A}$.
2: **while** True **do**
3:    Choose $S_0 \in \mathcal{S}, A_0 \in \mathcal{A}$ s.t. all pairs are chosen infinitely often.
4:    Generate an episode following $\pi$: $S_0, A_0, S_1, A_1, \ldots, S_{T-1}, A_{T-1}, S_T$.
5:    $G \leftarrow 0$
6:    **for** $t = T - 1, T - 2, \ldots, 0$ **do**
7:       $G \leftarrow G + r(S_t, A_t)$
8:       Append $G$ to $Returns(S_t, A_t)$
9:       $Q(S_t, A_t) \leftarrow average(Returns(S_t, A_t))$
10:      $\pi(S_t) \leftarrow \arg\max_a Q(S_t, a)$

---

We begin with the following lemma, whose proof can be found in Appendix A.

**Lemma 1.** *Let $X_1, X_2, \ldots$ be a sequence of random variables. Let $T$ be a random variable taking values in the positive integers, and suppose $P(T < \infty) = 1$. Suppose that for every positive integer $n$, $X_n, X_{n+1}, \ldots$ are i.i.d. with finite mean $x^*$ and finite variance when conditioned on $T = n$. Then $P(\lim_{N \to \infty} \frac{1}{N} \sum_{i=1}^{N} X_i = x^*) = 1$.*

Say that the MCES algorithm begins iteration $u$ at the $u$th time that an episode runs, and ends the iteration after $Q$ and $\pi$ have been updated (basically an iteration in the while loop in Algorithm 1). Denote by $Q_u(s, a)$ and $\pi_u(s)$ for the values of $Q(s, a)$ and $\pi(s)$ at the end of the $u$th iteration. For simplicity, assume that the MDP has a unique optimal policy $\pi^*$, and denote $q^*(s, a)$ for the corresponding action-value function. Note that the proof extends easily to the case of multiple optimal policies.

We now state and prove the convergence result for SFF MDPs. We provide the full proof here in order to highlight how simple it is. Afterwards we will state the more general result for OPFF MDPs, for which the proof is a little more complicated,

**Theorem 1.** *Suppose the MDP is SFF. Then $Q_u(s, a)$ converges to $q^*(s, a)$ and $\pi_u(s)$ converges to $\pi^*(s)$ for all $s \in \mathcal{S}$ and all $a \in \mathcal{A}$ w.p.1.*

*Proof.* Because the MDP is SFF, its MDP graph is a Directed Acyclic Graph (DAG). So we can re-order the $N$ states such that from state $s_k$ and selecting any action, we can only transition to a state in $\{s_{k+1}, \ldots, s_N\}$. Note state $s_N$ is the terminal state, from state $s_{N-1}$, all actions lead to $s_N$.

The proof is by backward induction. The result is trivially true for $s = s_N$. Suppose it is true for all states in $\{s_{k+1}, \ldots, s_N\}$ and all actions $a \in \mathcal{A}$. We now show it is true for $s_k$ and all actions $a \in \mathcal{A}$.

We first establish $Q_u(s_k, a)$ converges to $q^*(s_k, a)$ for all $a \in \mathcal{A}$ w.p.1. Let $T$ be the iteration $u$ when $\pi_u(s)$ has converged to $\pi_*(s)$ for all $s \in \{s_{k+1}, \ldots, s_N\}$. By the inductive assumption, $P(T < \infty) = 1$. Let $a \in \mathcal{A}$ be any action. Now consider any episode after time $T$ in which we visit state $s_k$ and choose action $a$. Because the MDP is SFF, the next state will be in $\{s_{k+1}, \ldots, s_N\}$, and because of the inductive assumption, the subsequent actions in the episode will follow the optimal policy $\pi^*$ until the episode terminates. By the definition of $q^*(s_k, a)$, the expected return for such an episode is equal to $q^*(s_k, a)$. Let $G_n$ denote the return for $(s_k, a)$ for the $n$th episode in which $(s_k, a)$ appears. After time $T$, these returns are i.i.d. with mean $q^*(s_k, a)$. Therefore, by Lemma 1,

$$\lim_{u \to \infty} Q_u(s_k, a) = \lim_{N \to \infty} \frac{1}{N} \sum_{n=1}^{N} G_n = q^*(s_k, a) \quad \text{w.p.1} \tag{4}$$

It remains to show that $\pi_u(s_k)$ converges to $\pi^*(s_k)$ w.p.1. Define $a^* = \pi^*(s_k)$. Since $\pi^*$ is the unique optimal policy, we have:

$$q^*(s_k, a^*) \geq q^*(s_k, a) + \epsilon' \tag{5}$$

for some $\epsilon' > 0$ for all $a \neq a^*$.

Let $\Omega$ be the underlying sample space, and let $\Lambda$ be the set of all $\omega \in \Omega$ such that $Q_u(s_k, a)(\omega)$ converges to $q^*(s_k, a)$ for all $a \in \mathcal{A}$. By (4), $P(\Lambda) = 1$. Thus for any $\omega \in \Lambda$ and any $\epsilon > 0$, there exists a $u'$ (depending on $\omega$) such that $u \geq u'$ implies

$$|q^*(s_k, a) - Q_u(s_k, a)(\omega)| \leq \epsilon \quad \text{for all } a \in \mathcal{A} \tag{6}$$

Let $\epsilon$ be any number satisfying $0 < \epsilon < \epsilon'/2$, let $\omega \in \Lambda$, and $u'$ be such that (6) is satisfied for all $u \geq u'$. It follows from (5) and (6) that for any $u \geq u'$ we have

$$Q_u(s_k, a^*)(\omega) \geq q^*(s_k, a^*) - \epsilon \tag{7}$$

$$\geq q^*(s_k, a) + \epsilon' - \epsilon \tag{8}$$

$$\geq Q_u(s_k, a)(\omega) + \epsilon' - 2\epsilon \tag{9}$$

$$> Q_u(s_k, a)(\omega) \tag{10}$$

for all $a \neq a^*$. Let $u$ be any iteration after $u'$ such that state $s_k$ is visited in the corresponding episode. From the MCES algorithm, $\pi_u(s_k)(\omega) = \arg\max_a Q_u(s_k, a)(\omega)$. Thus the above inequality implies $\pi_u(s_k)(\omega) = a^*$; furthermore, $\pi_u(s_k)(\omega)$ will be unchanged in any subsequent iteration. Thus, for every $\omega \in \Lambda$, $\pi_u(s_k)(\omega)$ converges to $a^*$. Since $P(\Lambda) = 1$, it follows $\pi_u(s_k)$ converges to $a^*$ w.p.1., completing the proof.

$\square$

We make the following additional observations: (1) It is not necessary to assume the MDP has a unique optimal policy. The theorem statement and proof go through with minor modification without this assumption. (2) Also we can allow for random reward, with distribution depending on the current state and action. The proof goes through with minor changes.

## 5 OPFF MDPs

For the case of OPFF, we first need to modify the algorithm slightly to address the issue that an episode might never reach the terminal state for some policies (for example, due to cycling). To this end, let $M$ be some upper bound on the number of states in our MDP. We assume that the algorithm designer has access to such an upper bound (which may be very loose). We modify the algorithm so that if an episode does not terminate within $M$ steps, the episode will be forced to terminate and the returns will simply not be used. We also initialize all Q values to be $-\infty$, so that the policy will always prefer an action that has led to at least one valid return over actions that never yield a valid return. The modified algorithm is given in Algorithm 2. Finally, as in Sutton & Barto (2018), we use first-visit returns for calculating the average return. (This mechanism is not needed for SFF MDPs since under all policies states are never revisited.)

---

**Algorithm 2** First-visit MCES for OPFF MDPs

---

1: Initialize: $\pi(s) \in \mathcal{A}$, $Q(s,a) = -\infty$, for all $s \in \mathcal{S}, a \in \mathcal{A}$, arbitrarily;
   $Returns(s,a) \leftarrow$ empty list, for all $s \in \mathcal{S}, a \in \mathcal{A}$.
2: **while** True **do**
3:    Choose $S_0 \in \mathcal{S}$, $A_0 \in \mathcal{A}$ s.t. all pairs are chosen infinitely often.
4:    Generate an episode following $\pi$: $S_0, A_0, S_1, A_1, \ldots, S_{T-1}, A_{T-1}, S_T$.
5:    **if** the episode does not end in less than $M$ time steps **then**
6:        terminate the episode at time step $M$
7:    **else**
8:        $G \leftarrow 0$
9:        **for** $t = T-1, T-2, \ldots, 0$ **do**
10:           $G \leftarrow G + r(S_t, A_t)$
11:           **if** $S_t, A_t$ does not appears in $S_0, A_0, S_1, A_1 \ldots, S_{t-1}, A_{t-1}$ **then**
12:               Append $G$ to $Returns(S_t, A_t)$
13:               $Q(S_t, A_t) \leftarrow average(Returns(S_t, A_t))$
14:               $\pi(S_t) \leftarrow \arg\max_a Q(S_t, a)$

---

**Theorem 2.** *Suppose the MDP is OPFF. Then $Q_u(s,a)$ converges to $q^*(s,a)$ and $\pi_u(s)$ converges to $\pi^*(s)$ for all $s \in \mathcal{S}$ and all $a \in \mathcal{A}$ w.p.1.*

The proof can be found in the appendix. As with SFF MDPs, it is not necessary to assume that the MDP has a unique optimal policy; furthermore, the reward can be random (with distribution depending on current state and action). Note for OPFF MDPs, our proof has to take a more sophisticated approach since we now can transition to arbitrary state by taking any non-optimal action.

## 6 FINITE-HORIZON MDPs

In this section, we extend our results to finite-horizon MDPs. In this case, we will be able to establish convergence for *all* MDPs (i.e., not just for OPFF MDPs). A finite-horizon MDP is defined by a finite horizon set $\mathcal{H} = \{0, 1, \ldots, H\}$, a finite state space $\mathcal{S}$, a finite action space $\mathcal{A}$, a reward function $r(s,t,a)$ mapping $\mathcal{S} \times \mathcal{H} \times \mathcal{A}$ to the reals, and a dynamics function $p(\cdot|s,t,a)$, which for every $s \in \mathcal{S}$, $a \in \mathcal{A}$ and time step $t$, gives a probability distribution over the state space $\mathcal{S}$. The horizon $H$ is the fixed time at which the episode terminates. Note in this setting, the optimal policy $\pi^*(s,t)$ will also be time-dependent, even if $p(\cdot|s,t,a)$ and $r(s,t,a)$ do not depend on $t$.

The MCES algorithm for this setting is given in Algorithm 3. Note that in this version of the algorithm, during exploring starts we need to choose an $S_0 \in \mathcal{S}$, $A_0 \in \mathcal{A}$, and an $h \in [0, H]$,.

**Corollary 1.** *Suppose we are using the finite-horizon optimization criterion. Then $Q_u(s,t,a)$ converges to $q^*(s,t,a)$ and $\pi_u(s,t)$ converges to $\pi^*(s,t)$ for all $s \in \mathcal{S}$, $t \in \mathcal{H}$ and all $a \in \mathcal{A}$ w.p.1.*

*Proof.* A finite-horizon MDP is equivalent to a standard MDP where the time step is treated as part of the state. Since the number of time steps monotonically increases, this MDP is SFF. Thus the convergence of MCES for the finite-horizon MDP setting follows directly from Theorem 1. $\square$

---

**Algorithm 3** MCES for Finite-Horizon MDPs

---

 1: Initialize: $\pi(s,t) \in \mathcal{A}$, $Q(s,t,a) \in \mathbb{R}$, for all $s \in \mathcal{S}, t \in [0,H], a \in \mathcal{A}$, arbitrarily;
    $Returns(s,t,a) \leftarrow$ empty list, for all $s \in \mathcal{S}, t \in [0,H], a \in \mathcal{A}$.
 2: **while** True **do**
 3:     Choose $S_0 \in \mathcal{S}, A_0 \in \mathcal{A}, h \in [0,H]$, s.t. all triples are chosen infinitely often.
 4:     Generate an episode following $\pi$: $S_h, A_h, S_{h+1}, A_{h+1}, \ldots, S_{H-1}, A_{H-1}, S_H$.
 5:     $G \leftarrow 0$
 6:     **for** $t = H-1, H-2, \ldots, h$ **do**
 7:         $G \leftarrow G + r(S_t, t, A_t)$
 8:         Append $G$ to $Returns(S_t, t, A_t)$
 9:         $Q(S_t, t, A_t) \leftarrow average(Returns(S_t, t, A_t))$
10:         $\pi(S_t, t) \leftarrow \arg\max_a Q(S_t, t, a)$

---

## 7 EXPERIMENTAL RESULTS

In addition to the theoretical results, we also provide experimental results to compare the convergence rate of the original MCES algorithm and the modified MCES algorithm in Tsitsiklis (2002), where the Q-function estimate is updated with Monte Carlo return only for the initial state-action pair of each episode. We will call the original MCES the "multi-update" variant, and the modified MCES the "first-update" variant. We consider two classical environments: blackjack and stochastic cliffwalking. We will briefly discuss the settings and summarize the results. A more detailed discussion, including additional results on Q-Learning is provided in appendix D and E.

For blackjack, we use the same problem setting as discussed in Sutton & Barto (1998). Blackjack is an episodic task where for each episode, the player is given 2 initial cards, and can request more cards (hits) or stops (sticks). The goal is to obtain cards whose numerical values sum to a number as great as possible without exceeding 21. Along with the two MCES variants, we also compare two initialization schemes: standard-init, where we initialize by first drawing two random cards from a deck of 52 cards; and uniform-init, where the initial state for an episode is uniformly sampled from the state space. We now compare their rate of convergence in terms of performance and the absolute Q update error, averaged across state-action pairs visited in each episode. Figure 1 shows uniform initialization converges faster than standard initialization, and the multi-update variant is faster than the first-update variant for both initialization schemes.

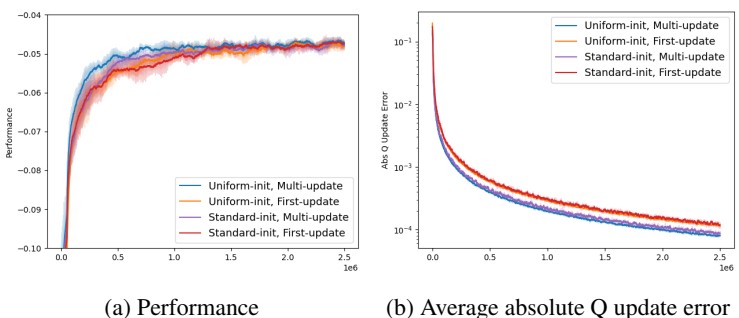

(a) Performance        (b) Average absolute Q update error

Figure 1: (a) Performance and (b) average of absolute Q update error (in log scale) for blackjack. Multi-update variant has better performance and lower Q error.

In cliff walking, an agent starts from the bottom left corner of a gridworld, and tries to move to the bottom right corner, without falling into a cliff (an area covering the bottom row of the gridworld). The agent gets a -1 reward for each step it takes, and a -100 for falling into the cliff. We consider two variants: SFF cliff walking and OPFF cliff walking tasks. For the SFF variant, the current time step of the episode is part of the state, so the MDP is a SFF MDP, and the agent tries to learn a time-dependent optimal policy. In this setting we add more stochasticity by having wind move the agent towards one of four directions with some probability. For the OPFF variant, the time step is not part of the state, and the wind only affect the agent when the agent moves to the right, and can only blow the agent one step upwards or downwards, making an OPFF MDP. We now compare the two MCES

variants and measure the convergence rate in terms of the performance and the average L1 distance between the current Q estimate and the optimal Q values. The results are summarized in Figure 2, the training curves are averaged across a number of different gridworld sizes and wind probabilities. Results show that multi-update MCES consistently learns faster than first-update MCES.

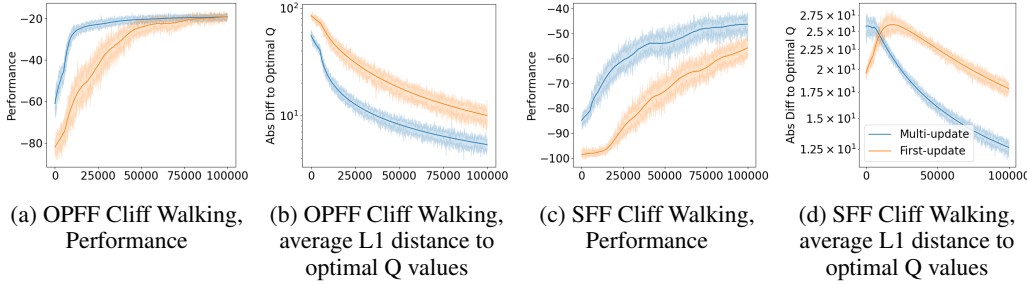

(a) OPFF Cliff Walking, Performance

(b) OPFF Cliff Walking, average L1 distance to optimal Q values

(c) SFF Cliff Walking, Performance

(d) SFF Cliff Walking, average L1 distance to optimal Q values

Figure 2: Performance and the average L1 distance to optimal Q values, on OPFF and SFF cliff walking, the curves are averaged over different gridworld sizes and wind probability settings.

Our experimental results show that the multi-update MCES converges much faster than the first-update MCES. These results further emphasize the importance of gaining a better understanding of the convergence properties of the original multi-update MCES variant, which is much more efficient.

## 8  CONCLUSION

Theorem 2 of this paper shows that as long as the episodic MDP is OPFF, then the MCES algorithm converges to the optimal policy. As discussed in Section 3.2, many environments of practical interest are OPFF. Our proof does not require that the Q-values be updated at the same rate for all state-action pairs, thereby allowing more flexibility in the algorithm design. Our proof methodology is also novel, and can potentially be applied to other classes of RL problems. Moreover, our methodology also allows us to establish convergence results for finite-horizon MDPs as a simple corollary of Theorem 2. Combining the results of Bertsekas & Tsitsiklis (1996), Tsitsiklis (2002), Chen (2018), Liu (2020), and the results here gives Figure 3, which summarizes what is now known about convergence of the MCES algorithm. In appendix F, we also provide a new counterexample where we prove that MCES may not converge in a non-OPFF episodic environment.

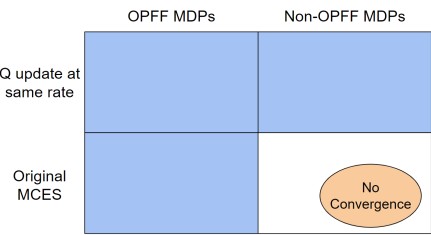

Figure 3: MCES has been studied for two classes of algorithms: Q-values updated at the same rate Tsitsiklis (2002); Chen (2018); Liu (2020) and the original, more flexible algorithm, which does not require the conditions $(ii)$ and $(iii)$ stated in the Introduction. We partition the episodic MDP space into two classes: OPFF MDPs and non-OPFF MDPs. As shown in the figure, this leads to four algorithmic/MDP regions. Convergence is now established for the three blue shaded regions, for all discount factors $\gamma \leq 1$. In the other region, it is known that at least for some non-OPFF MDPs, convergence does not occur (shown as the orange oval region). A new counterexample in episodic MDP and additional discussion are provided in appendix F.

The results in this paper along with other previous works (Tsitsiklis, 2002; Chen, 2018; Liu, 2020) make significant progress in establishing the convergence of the MCES algorithm. Many cases of practical interest are covered by the conditions in these papers. It still remains an open problem whether there exist conditions on MDPs that are weaker than the OPFF, and can still guarantee the convergence of the original MCES algorithm.

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

## A  PROOF OF LEMMA 1

*Proof.* We have

$$P(\lim_{N \to \infty} \frac{1}{N} \sum_{i=1}^{N} X_i = x^*)$$

$$= \sum_{n=1}^{\infty} P(\lim_{N \to \infty} \frac{1}{N} \sum_{i=1}^{N} X_i = x^* | T = n) P(T = n)$$

$$= \sum_{n=1}^{\infty} P(\lim_{N \to \infty} (\frac{1}{N} \sum_{i=1}^{n-1} X_i + \frac{1}{N} \sum_{i=n}^{N} X_i) = x^* | T = n) P(T = n)$$

$$= \sum_{n=1}^{\infty} P(\lim_{N \to \infty} \frac{1}{N} \sum_{i=n}^{N} X_i = x^* | T = n) P(T = n) = 1$$

The first equality follows from $P(T < \infty) = 1$. The last equality follows from the conditional i.i.d. assumption and the Strong Law of Large Numbers. Note we can apply the Strong Law of Large Numbers here because as stated in Lemma 1, we have the assumption that $X_n, X_{n+1}, \ldots$ are i.i.d. with finite mean and finite variance.

$\square$

## B  PROOF OF THEOREM 2

*Proof.* Because the MDP is OPFF, its optimal policy MDP graph is a DAG, so we can re-order the states such that from state $s_k$ and selecting the optimal action $a_k^*$, we can only transition to a state in $\{s_{k+1}, \ldots, s_N\}$. Note that state $s_N$ is the terminal state, and note that from state $s_{N-1}$ the optimal action only leads to $s_N$.

We first show that for all $k = 1, \ldots, N$, $Q_u(s_k, a_k^*)$ converges to $q^*(s_k, a_k^*)$ and that $\pi_u(s_k)$ converges to $\pi^*(s_k)$, w.p.1. Once we establish this convergence result for all $k = 1, \ldots, N$, we will then complete the proof by establishing convergence of $Q_u(s_k, a)$ for arbitrary actions $a$. Note that the organization of this proof is slightly more complicated than that of Theorem 1 due to the weaker OPFF assumption.

The result is trivially true for $s = s_N$. Suppose now $Q_u(s_j, a_j^*)$ converges to $q^*(s_j, a_j^*)$ and that $\pi_u(s_j)$ converges to $\pi^*(s_j)$, w.p.1 for all $j = k+1, \ldots, N$. We now show $Q_u(s_k, a_k^*)$ converges to $q^*(s_k, a_k^*)$ and that $\pi_u(s_k)$ converges to $\pi^*(s_k)$, w.p.1. Denote $a^* = a_k^*$.

Let $T$ be the iteration $u$ when $\pi_u(s)$ has converged to $\pi_u^*(s)$ for all $s \in \{s_{k+1}, \ldots, s_N\}$. By the inductive assumption, $P(T < \infty) = 1$. Now consider any episode after time $T$ that for some timestep in this episode, we arrive in state $s_k$ and chooses the optimal action $a^*$. Because the MDP is OPFF, the next state will be in $\{s_{k+1}, \ldots, s_N\}$, and because of the inductive assumption the subsequent actions in the episode will follow the optimal policy $\pi^*$ until the episode terminates within a finite number of steps. By the definition of $q^*(s_k, a^*)$, the expected return for $(s_k, a^*)$ in this episode is equal to $q^*(s_k, a^*)$. Let $G_n$ denote the return for $n$th episode in which we visit $s_k$ and chooses optimal action $a^*$. Note that after time $T$, these returns are i.i.d. with mean $q^*(s_k, a^*)$. Therefore, by Lemma 1,

$$\lim_{u \to \infty} Q_u(s_k, a^*) = \lim_{N \to \infty} \frac{1}{N} \sum_{n=1}^{N} G_n = q^*(s_k, a^*) \quad \text{w.p.1} \tag{11}$$

Next we show that $\pi_u(s_k)$ converges to $\pi^*(s_k) = a^*$ w.p.1. Since $\pi^*$ is the unique optimal policy, we have:

$$q^*(s_k, a^*) \geq q^*(s_k, a) + \epsilon' \tag{12}$$

for some $\epsilon' > 0$ for all $a \neq a^*$.

The proof at this stage is different from the proof at the corresponding stage in the proof of Theorem 1 since we can now only use (11) for $a = a^*$.

Consider state $s_k$ and an arbitrary action $a \in \mathcal{A}$. From the MCES algorithm, we have $Q_u(s_k, a) = \frac{1}{L_u} \sum_{l=1}^{L_u} G_l$, where $L_u$ is the total number of first-visit returns used to compute $Q_u(s_k, a)$ up through the $u$th iteration, and $G_l$ is the return value for the $l$th such iteration. Let $\Pi$ denote the (finite) set of all deterministic policies, $L_u^\pi$ denote the number of first-visit returns used to compute $Q_u(s_k, a)$ up through the $u$th iteration when using policy $\pi$, and $G_l^\pi$, $1 \leq l \leq L_u^\pi$, denote the $l$th return value when policy $\pi$ is used. We have

$$Q_u(s_k, a) = \frac{1}{L_u} \sum_{l=1}^{L_u} G_l \tag{13}$$

$$= \sum_{\pi \in \Pi} \frac{L_u^\pi}{L_u} \left( \frac{1}{L_u^\pi} \sum_{l=1}^{L_u^\pi} G_l^\pi \right) \tag{14}$$

By the law of large numbers, we know that for any policy $\pi$ such that $L_u^\pi \to \infty$ we have w.p.1

$$\lim_{u \to \infty} \frac{1}{L_u^\pi} \sum_{l=1}^{L_u^\pi} G_l^\pi = q^\pi(s_k, a) \leq q^*(s_k, a) \tag{15}$$

where $q^\pi(s_k, a)$ is the action-value function for policy $\pi$. The inequality in (15) follows from the definition of $q^*(s, a)$. It follows from (14) and (15) that w.p.1

$$\limsup_{u \to \infty} Q_u(s_k, a) \leq q^*(s_k, a) \quad \text{for all } a \tag{16}$$

Note that in the OPFF setting, in the special case that no valid return has been obtained for $(s_k, a)$, then $Q_u(s_k, a) = -\infty \leq q^*(s_k, a)$, so the above inequality still holds.

Let $\Omega$ be the underlying sample space, and let $\Lambda \subset \Omega$ be the set over which (16) holds. Note that $P(\Lambda) = 1$. Thus for any $\omega \in \Lambda$ and any $\epsilon > 0$, there exists a $u'$ such that $u \geq u'$ implies

$$Q_u(s_k, a)(\omega) \leq q^*(s_k, a) + \epsilon \quad \text{for all } a \in \mathcal{A} \tag{17}$$

Let $\epsilon$ be any number satisfying $0 < \epsilon < \epsilon'/2$, let $\omega \in \Lambda$, and $u'$ be such that (17) is satisfied for all $u \geq u'$. It follows from (11), (12) and (17) that for any $u \geq u'$ we have

$$Q_u(s_k, a^*)(\omega) \geq q^*(s_k, a^*) - \epsilon \tag{18}$$

$$\geq q^*(s_k, a) + \epsilon' - \epsilon \tag{19}$$

$$\geq Q_u(s_k, a)(\omega) + \epsilon' - 2\epsilon \tag{20}$$

$$> Q_u(s_k, a)(\omega) \tag{21}$$

for all $a \neq a^*$. Let $u$ be any iteration after $u'$ such that the episode includes state $s_k$. From the MCES algorithm, $\pi_u(s_k)(\omega) = \arg\max_a Q_u(s_k, a)(\omega)$. Thus the above inequality implies $\pi_u(s_k)(\omega) = a^*$; furthermore, $\pi_u(s_k)(\omega)$ will be unchanged in any subsequent iteration. Thus, for every $\omega \in \Lambda$, $\pi_u(s_k)(\omega)$ converges to $a^*$. Since $P(\Lambda) = 1$, it follows $\pi_u(s_k)$ converges to $a^*$ w.p.1.

We have now shown that for all $k = 1, \ldots, N$, $Q_u(s_k, a_k^*)$ converges to $q^*(s_k, a_k^*)$ and that $\pi_u(s_k)$ converges to $\pi^*(s_k)$, w.p.1. It remains to show convergence of $Q_u(s, a)$ to $q^*(s, a)$ for arbitrary state $s$ and action $a$. Let $u'$ be such that $u \geq u'$, $\pi_u(s) = \pi^*(s)$ for all $s \in \mathcal{S}$. Consider an episode after iteration $u'$ in which we visit state $s$ and take action $a$. After taking action $a$, the policy follows the optimal policy $\pi^*$. Thus the expected return for $(s, a)$ in this episode is $q^*(s, a)$. We can thus once again apply Lemma 1 to show $Q_u(s, a)$ converges to $q^*(s, a)$ w.p.1, thereby completing the proof. $\square$

## C  ADDITIONAL DISCUSSION ON PRACTICAL OPFF TASKS AND MC ALGORITHMS

### C.1  OPFF AS A PRACTICAL SETTING

OPFF MDP is a large and important family of MDPs. There are many natural examples of OPFF MDPs, for example, Blackjack, windy gridworlds and MuJoCo robotic locomotion as discussed in this work. Also note that if a task involves any monotonically changing value as part of the state, then it is also OPFF. For example, when time is added to the state to handle finite horizon criteria, then the MDP becomes OPFF whether or not the original MDP is OPFF. Here we give an extended list of real-world practical problems that fall into OPFF MDPs:

- Operating a robot or datacenter with a power budget;
- Driving a car with a given amount of fuel or to reach a target within a time limit;
- Manufacturing a product with limited resources;
- Doing online ads bidding with a fixed budget;
- Running a recommendation systems with a limited amount of recommendation attempts;
- Trading to maximize profit within a time period, and more;

For the MuJoCo environment, note if we treat it as an episodic MDP (for example, if the task is to run towards a goal, and the episode terminates when the goal is reached) instead of an infinite-horizon task (for example, if the task is to keep running indefinitely), then it falls into the category of OPFF because the simulation is deterministic. As discussed in the main paper, all deterministic episodic MDPs are OPFF.

### C.2  HOW ALPHAZERO RELATES TO MONTE CARLO METHODS

The AlphaZero learning process can be roughly seen as a nested loop: there is the outer loop where the agent starts the game from an empty board and plays to the end of the game, and then for each state $s$ on this trajectory, there is an inner loop of Monte Carlo Tree Search (MCTS), which is a planning phase that starts from state $s$. After the planning finishes, a single physical move is made (in the outer loop). When we consider all algorithmic components of AlphaZero, it is clear that AlphaZero is very different from MCES (even the MCTS algorithm alone is very different from MCES). Although MCES and AlphaZero do not do the same thing, they share some important similarities on a high level.

In appendix E, we further provide a discussion on the use of Monte Carlo methods in recent literature, together with an additional experimental comparison between the MCES algorithm and the Q-Learning algorithm.

# D   ADDITIONAL EXPERIMENTAL DETAILS

## D.1   BLACKJACK ENVIRONMENT

Here we give a more in-depth discussion on our blackjack experiments. The code for the experiments is provided [1].

We use the same problem setting as discussed in Sutton & Barto (1998). We consider only a single player against a dealer. The goal of the blackjack game is to request a number of cards so that the sum of the numerical values of your cards is as great as possible without exceeding 21. All face cards will be counted as 10 and an ace can be counted as either 1 or 11. If the player holds an ace that can be counted as 11 without causing the total value to exceed 21, then the ace is said to be usable.

For each round of blackjack, both the player and the dealer will get two cards respectively at the beginning. One of the dealer's cards is face up and another is face down. For each time step in the episode, the player has two possible actions: either to request an additional card (hits), or stops (sticks), if the player's sum exceeds 21 (goes bust), then the player loses. After the player sticks, The dealer will hit or stick according to a fixed policy: he sticks on any sum of 17 or greater and hits otherwise, if the dealer goes bust then the player wins. In the end, if both sides do not go bust, their sums are compared, and the player wins if the player's sum is greater, and loses if the dealer's sum is greater, and the game is a draw if both sides have the same sum. When the episode terminates, the reward is 1 if the player wins, -1 if the player loses, and 0 if the game is a draw. Note that in an episode, the player cannot revisit a previously visited state (if the player has a usable ace, the player can move into a state with a unusable ace, but not vice-versa). Therefore blackjack is a stochastic feed-forward (SFF) environment.

For blackjack, the player's state consists of three components: the player's current sum, the card that the dealer is showing, and whether the player has a usable ace. There are some states where the optimal policy is trivial. For example, when the player has a very small sum, the player can always hit and the sum value will increase for sure without the danger of going bust. If we ignore such special states, then we have a total of 200 states that are interesting to the player (it is 200 states considering the player's sum (12-21), dealer's showing card (ace-10) and whether the player has a usable ace (Sutton & Barto, 1998)). For the uniform initialization scheme, the initial state of the player is uniformly sampled from these 200 states.

For the standard initialization, we instead simply follow the rules of blackjack. We randomly draw cards for the player and the dealer, each of these cards can be one of the 13 cards (ace-10, J, Q and K) with equal probability. (The cards are dealt from an infinite deck, i.e., with replacement. )

When we compare the convergence rate of the multi-update MCES and the first-update MCES, we use performance and the average absolute Q update error. The Q update error for a state-action pair is simply the difference between the Q values before and after a Q update. For each episode, the average absolute Q update error is averaged over all state-action pairs that are updated in this episode. In all figures, the solid curve shows the mean value over 5 seeds, and the shaded area indicates the confidence interval.

In Figure 1, we see that the first-update MCES variant (which only updates the Q value for the initial state-action pair in each episode) works in blackjack even when we sample according to the standard rule of blackjack and not uniformly. This might be due to fact that for blackjack, although the initial state distribution under standard initialization (where the player gets 2 random initial cards from the dealer) is not the same as uniform initialization, it still sufficiently covers the state space, so the agent is able to update its value estimates for all state-action pairs and gradually learn a good policy. However, this is not the case for cliffwalking, where if we initialize an episode according to the standard rule, then the initial state for the agent is always at the the bottom left corner of the gridworld, making it impossible to update the value function for other states. In many practical problems, we might not have access to an initial state distribution that sufficiently covers the entire state space, so it is very important to gain a better understanding of the convergence properties of the multi-update MCES variant.

---

[1]https://github.com/Hanananah/On-the-Convergence-of-the-MCES-Algorithm-for-RL

### D.2 STOCHASTIC CLIFF WALKING ENVIRONMENT

Here we give a more in-depth discussion on our cliff walking experiments.

As illustrated in Figure 4, in the cliff walking environment, the agent starts from the bottom left corner of a gridworld (marked with the letter S), and tries to move to the goal state at the bottom right corner (marked with the letter G), without falling into the cliff area (shaded area at the bottom row of the gridworld).

In our setting, the agent can move in four directions (right, up, left, down), and gets a -1 reward for each step it takes, a -100 for falling into the cliff, and a 0 for reaching the goal state. If the agent reaches the goal state or falls off the cliff, the episode will immediately terminate. If the agent moves out of the boundary of the gridworld, the agent will be "bounced" back and will not be able to go outside the boundaries.

Here we consider two variants of the cliffwalking environment: SFF cliff walking and OPFF cliff walking. For the SFF variant, the current time step of the episode is part of the state. Since the time step can only increase, the MDP is now a SFF MDP, and the agent tries to learn a time-dependent optimal policy. In this setting we add stochasticity by having wind move the agent one step towards one of the four directions with some probability. For example, when the wind probability is 50%, there is 12.5% chance that the agent will take an additional random step for each direction due to the wind. In Figure 4 this is illustrated with the arrows near the position $P_2$. If the agent is in the SFF cliff walking environment and is currently at position $P_2$, then if the agent moves one step upward (indicated by the solid arrow), there is some chance that the agent will also take an additional step towards one of the four directions, with equal probability (indicated by dashed arrows). Note that the agent's state space now includes the $x, y$ positions of the agent, as well as the current time step.

For the OPFF variant, the time step is not part of the state. To make sure the environment is OPFF, we only allow the wind to affect the agent when it moves to the right, and the wind can only blow the agent one step upwards, or downwards. This is shown in Figure 4 by the arrows near the position $P_1$, if the agent is in the OPFF cliff walking environment and is current in position $P_1$, then if the agent moves one step to the right, the wind has some probability of making it take an additional step upwards or downwards, with equal probability. In this setting, if the agent takes any action other than moving one step right, then it will not be affected. Under such a setting, if the agent is following an optimal policy, then in an episode it will never revisit a previously visited state, thus the MDP is OPFF.

Special care is required when running MCES in the OPFF environment, since it is possible the agent can run into infinite loops (e.g. keep bumping into the boundary), depending on the initial policy. One simple method to tackle this issue is to have an artificial time limit, and if during one episode the agent reaches this time limit, the episode is terminated and the agent receives a large, negative reward. This allows deterministic agents to train effectively in these OPFF environments.

#### D.2.1 CLIFF WALKING EXPERIMENTAL RESULTS

We use performance and the average L1 distance to the optimal Q value as metrics for the convergence rates of the multi-update and first-update MCES variants. Here the optimal Q values are computed using value iteration. The full experimental results are shown in Figures 5, 6, 7 and 8. We present results on gridworld sizes of $8 \times 6, 12 \times 9, 16 \times 12$ and wind probability $= 0.1, 0.3, 0.5$.

In Fig 5 and Fig 6, we consider the OPFF cliff walking environment, where time is not included in the state space and the wind only affects the action to the right.

In Fig 7 and Fig 8, we consider the SFF cliff walking environment, where the current time step is included as part of the state and the wind can affect the agent in all four directions.

Results in both cliff walking environments, and across all tested gridworld size and wind probability consistently show that multi-update MCES learns faster than first-update MCES. These results show there is indeed a significant performance difference between these two variants of the MCES algorithm. We also observe a general trend that the performance gap tends to increase when the state space becomes larger (a larger gridworld) and when we have more stochasticity in the environment (a higher wind probability). In Figure 6 and 8, each row shows performance for a gridworld size and each column shows a wind probability, if we go from the top left figure (figure (a)) to the bottom

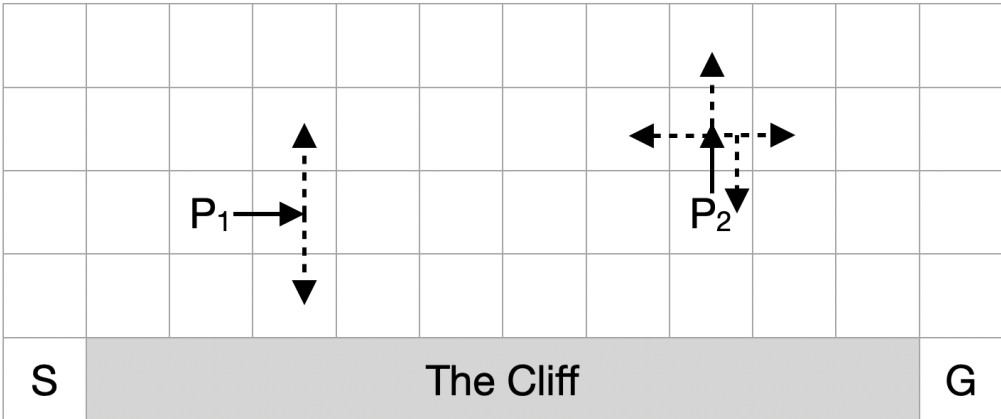

Figure 4: Cliff Walking Environment Illustration: The grid world contains of a start state, a goal state and a cliff area between them. In the OPFF cliff walking setting with wind probability $\mathbb{P}_w$, when the agent is at $P_1$ and moves one step to the right, it will take an additional step upwards or downwards, each with probability $\frac{\mathbb{P}_w}{2}$ and not take this additional step with probability $1 - \mathbb{P}_w$. In the SFF cliff walking environment with wind probability $\mathbb{P}_w$, when the agent is at $P_2$ and takes one step up, it will take an additional step towards one of the four directions, each with probability $\frac{\mathbb{P}_w}{4}$, and take no additional step with probability $1 - \mathbb{P}_w$.

right figure (figure (i)), we see that the performance gap tends to become bigger. This result shows that taking more updates to the value estimate can be important especially in complex tasks with more randomness.

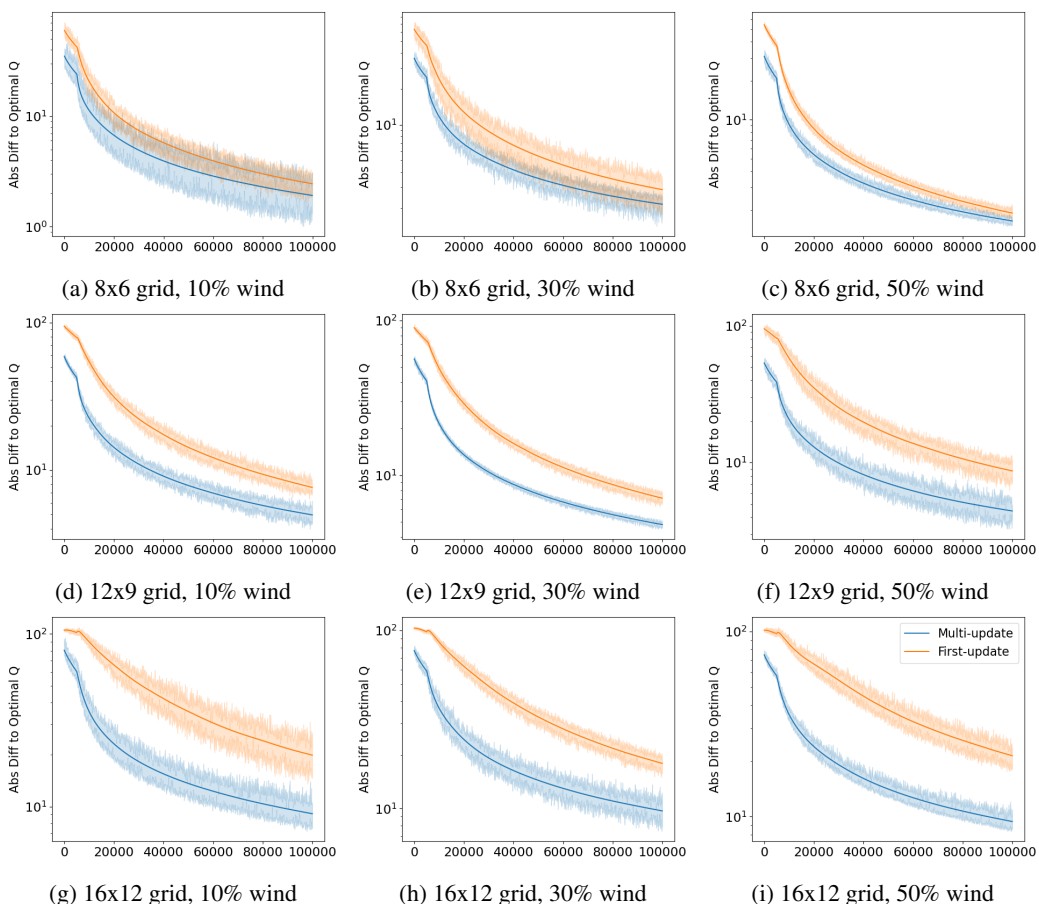

Figure 5: OPFF Cliff Walking: average L1 distance to optimal Q values, with different grid world sizes and wind probabilities. Y-axis is in log scale.

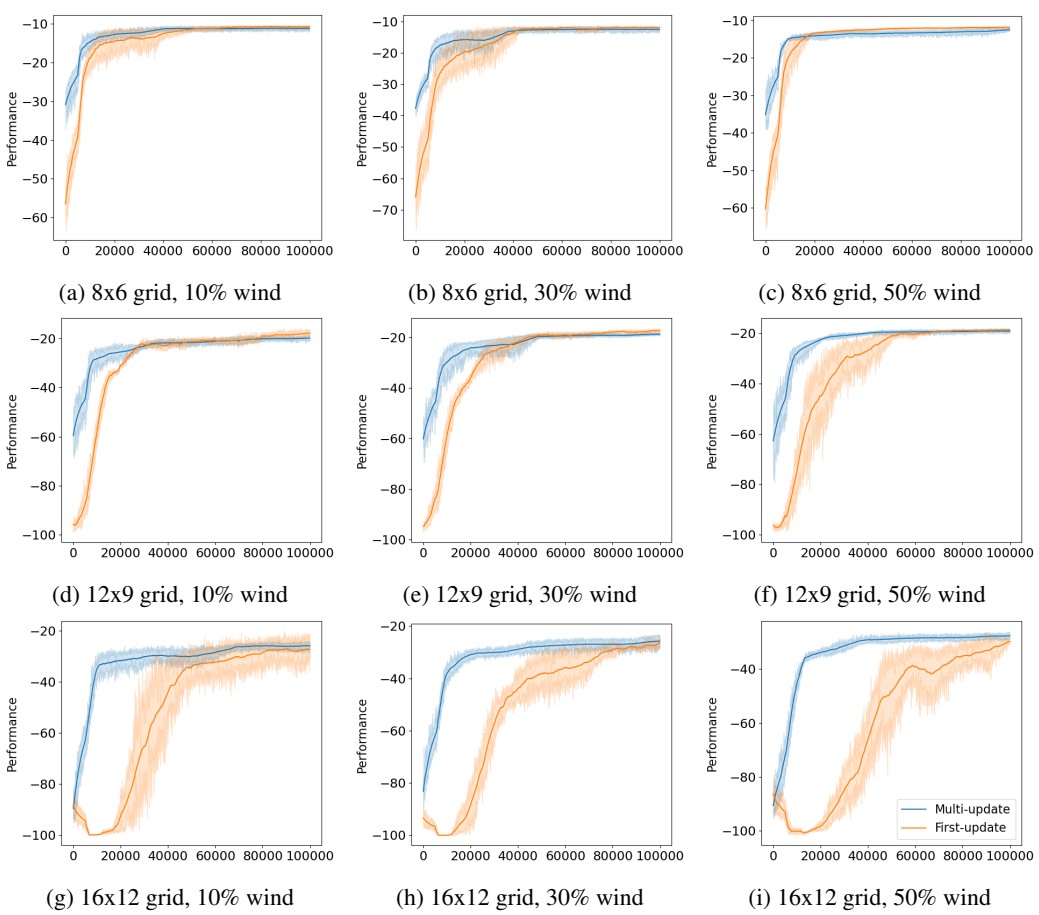

Figure 6: OPFF Cliff Walking: Performance, with different grid world sizes and wind probabilities.

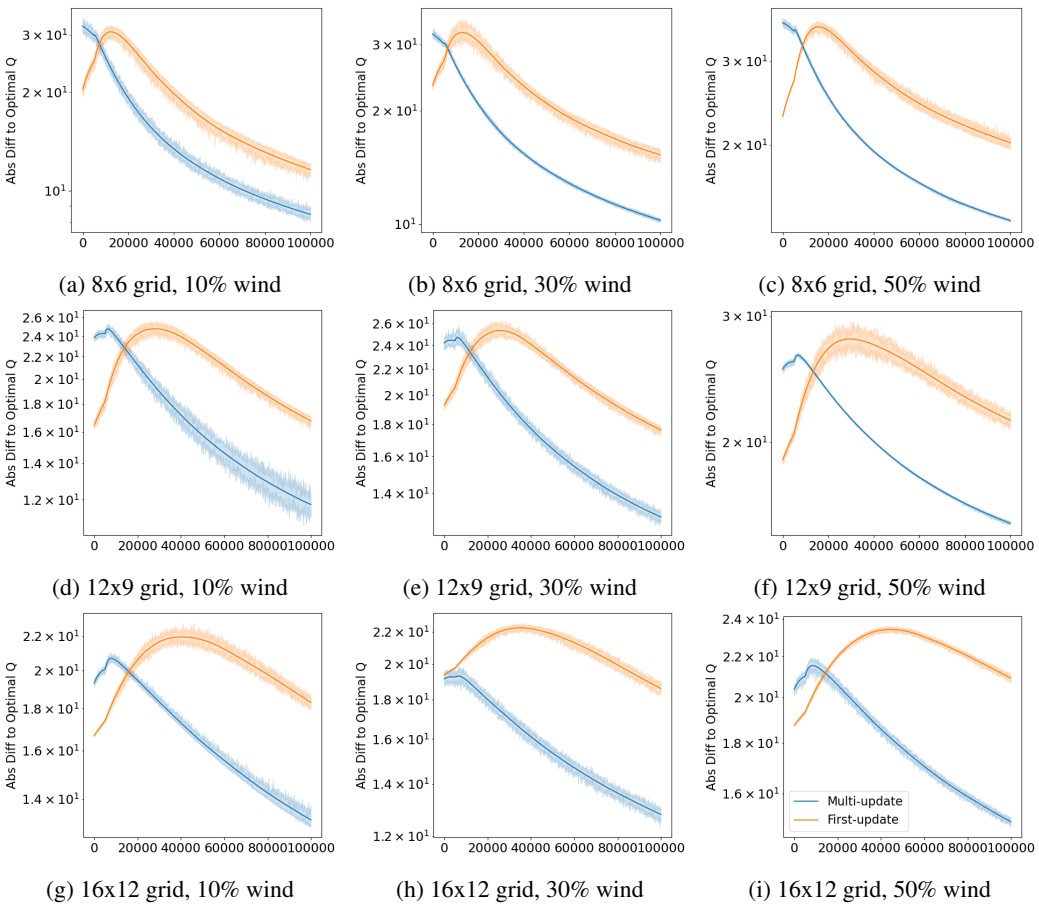

Figure 7: SFF Cliff Walking: average L1 distance to optimal Q values, with different grid world sizes and wind probabilities, y axis is in log scale.

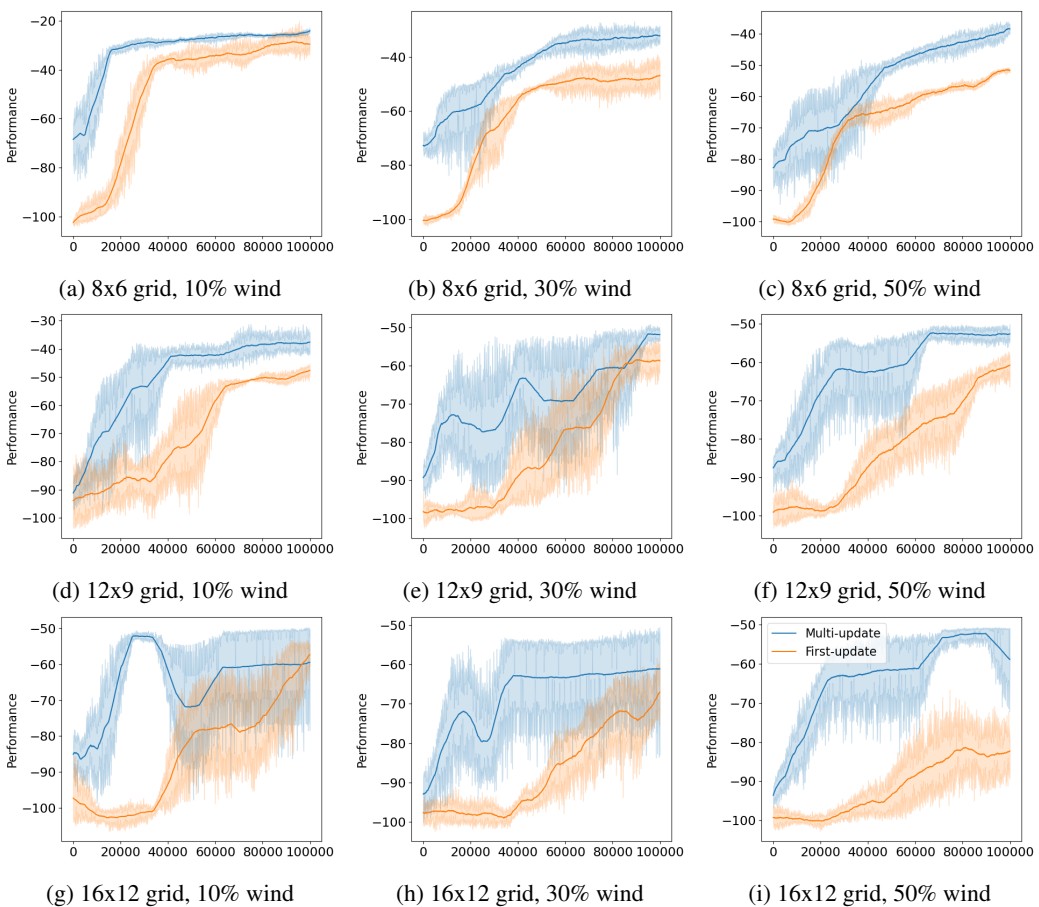

Figure 8: SFF Cliff Walking: Performance, with different grid world sizes and wind probabilities.

# E    ADDITIONAL EXPERIMENTS ON Q-LEARNING

qIn this section, we additionally compare the multi-update MCES variant to the Q-learning algorithm with different learning rates and discount factors. The results are summarized in Figure 9, the training curves are averaged across a number of different gridworld sizes and wind probabilities. Results show that in the CliffWalking environment, the optimal hyperparameters for Q-Learning are different in each environment. With the right set of hyperparameters, Q-Learning can significantly outperform the multi-update variant of MCES. This observation is consistent with the strong sample efficiency of a large number of Q-Learning-based methods in recent deep reinforcement learning literature. Note that some of the most effective deep reinforcement learning methods, such as DrQv2 (Yarats et al., 2021), use a variant of the n-step TD method, which can be seen as a generalization of both MC and one-step TD methods, as discussed in Chapter 7 of Sutton & Barto (2018). Popular on-policy deep reinforcement learning methods such as PPO (Schulman et al., 2017) also use a technique called generalized advantage estimation, which can also be seen as a mix of MC and TD methods. In offline deep reinforcement learning, some recent methods use Monte Carlo returns to select a number of "best actions" and perform imitation learning, effectively avoiding some of the unique issues in offline reinforcement learning (Chen et al., 2020).

These results show that it is important to improve our understanding of both MC and TD methods, in order to develop new algorithms that are efficient and performant. Detailed results for each setting are shown in Figure 10, Figure 11, Figure 12 and Figure 13.

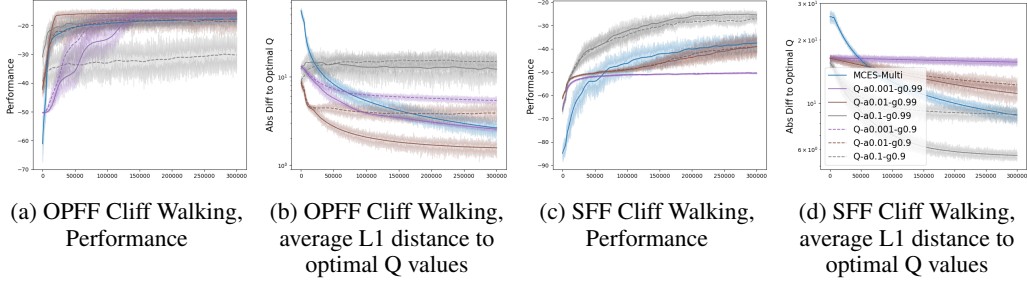

(a) OPFF Cliff Walking, Performance

(b) OPFF Cliff Walking, average L1 distance to optimal Q values

(c) SFF Cliff Walking, Performance

(d) SFF Cliff Walking, average L1 distance to optimal Q values

Figure 9: Performance and the average L1 distance to optimal Q values, on OPFF and SFF cliff walking, for MCES with multi-update variant and Q-Learning with different learning rate ($a$) and discount factor ($g$), the curves are averaged over different gridworld sizes and wind probability settings.

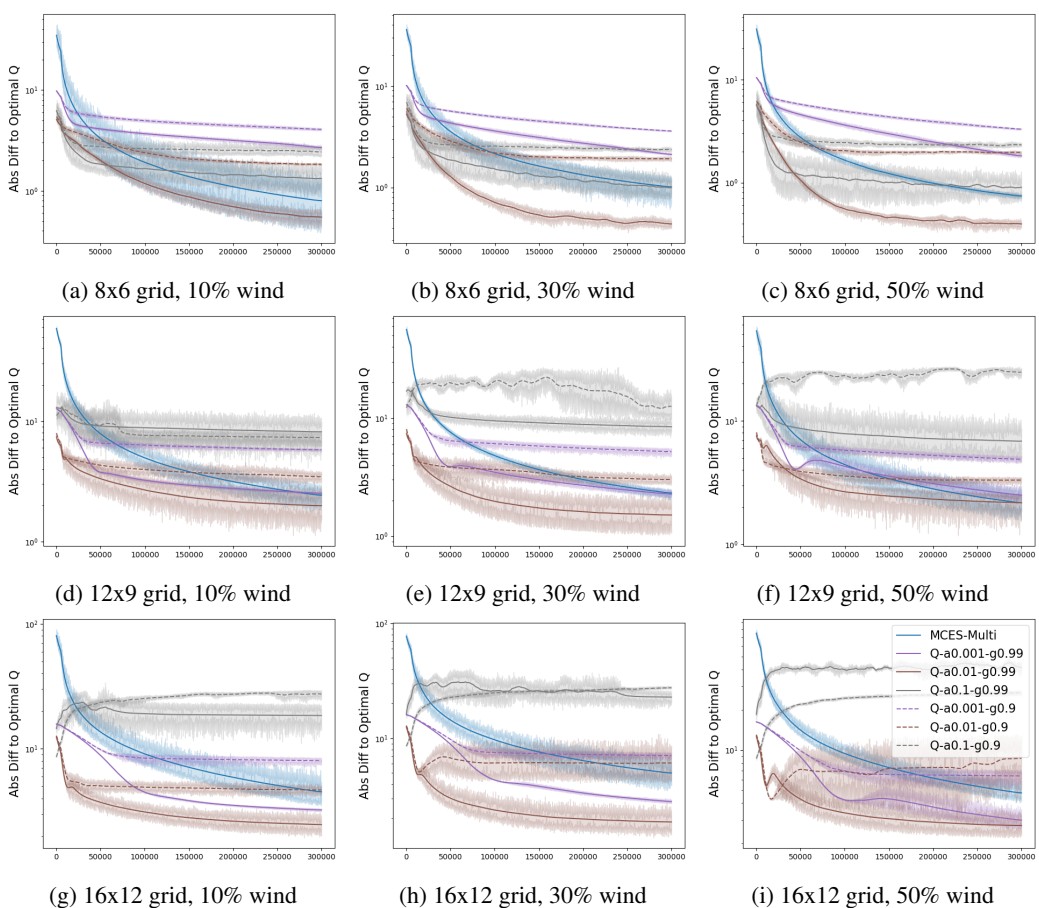

Figure 10: OPFF Cliff Walking: average L1 distance to optimal Q values, with different grid world sizes and wind probabilities. Y-axis is in log scale.

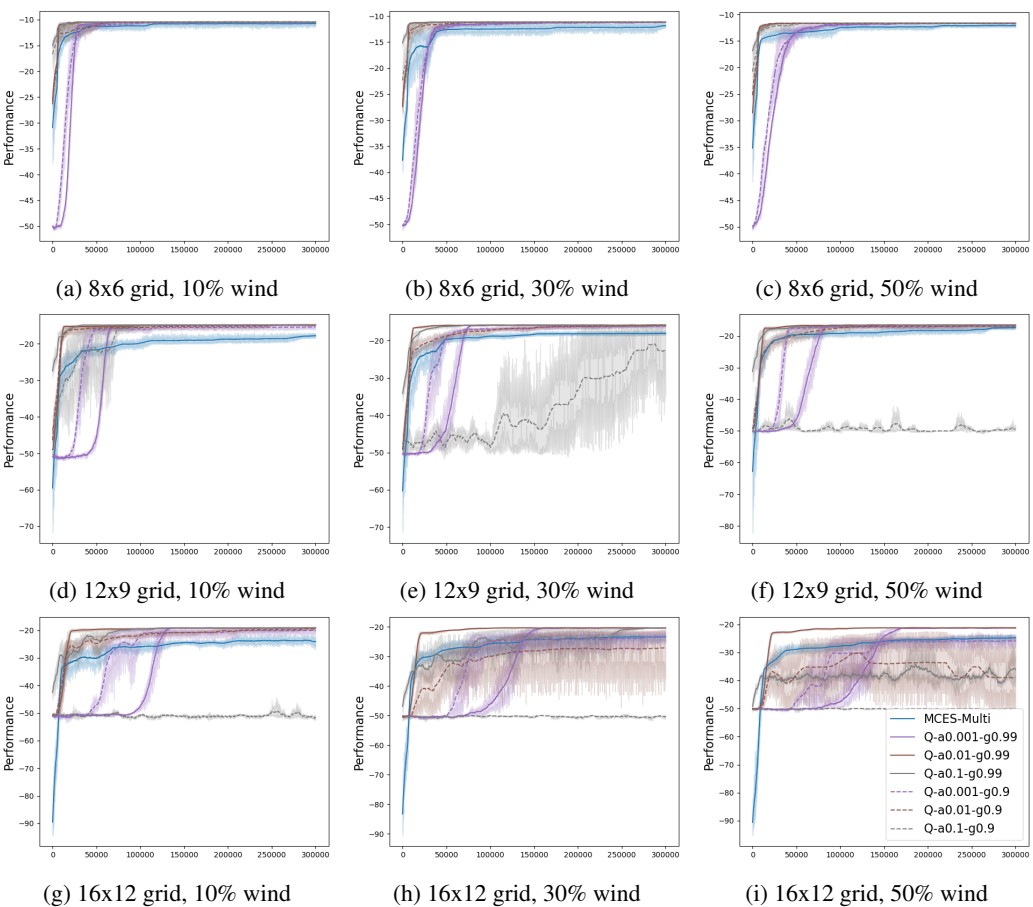

Figure 11: OPFF Cliff Walking: Performance, with different grid world sizes and wind probabilities.

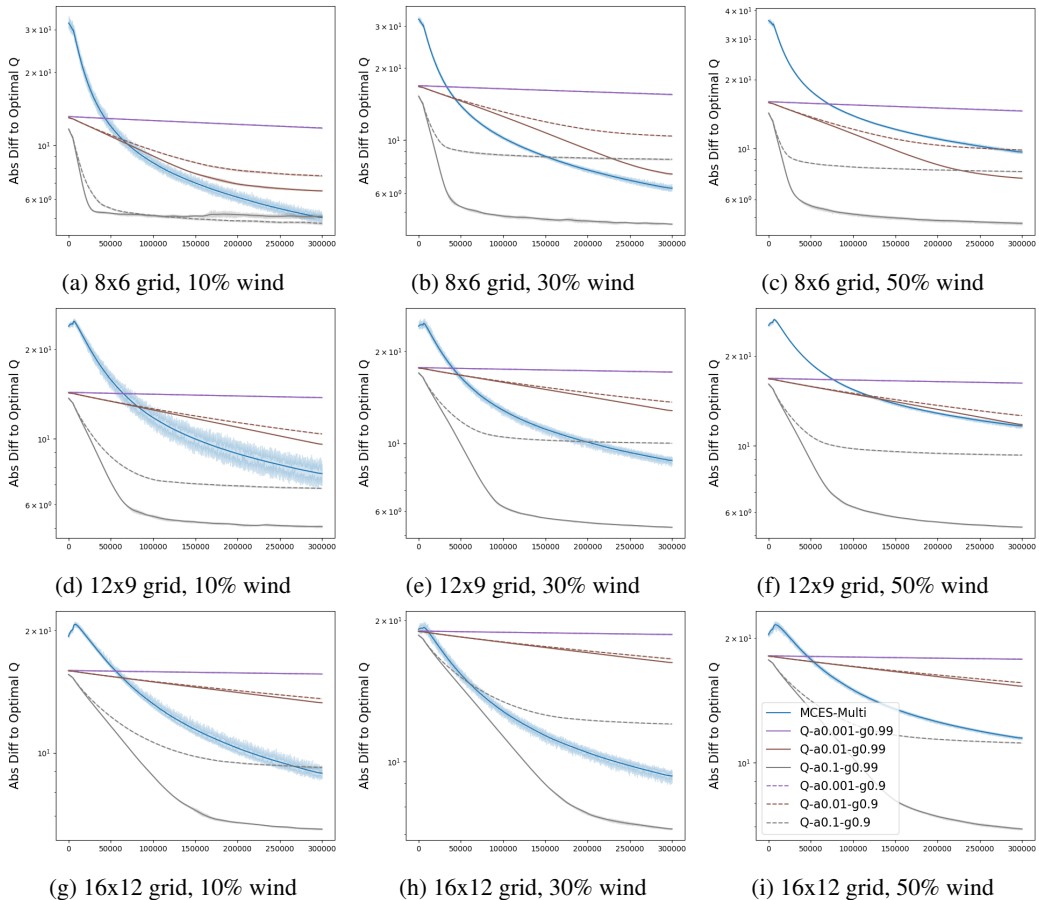

Figure 12: SFF Cliff Walking: average L1 distance to optimal Q values, with different grid world sizes and wind probabilities. Y-axis is in log scale.

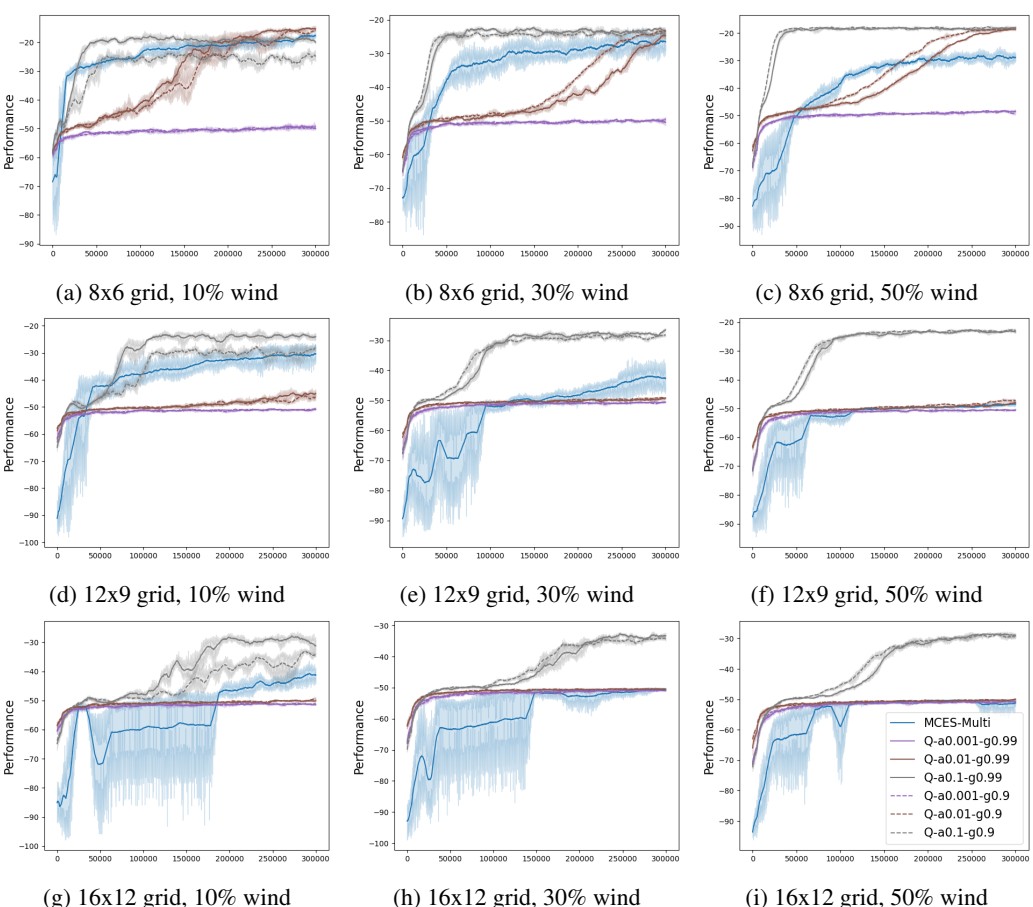

Figure 13: SFF Cliff Walking: Performance, with different grid world sizes and wind probabilities.

# F    COUNTEREXAMPLE

In this section, we present a counterexample in which the MCES algorithm (4) fails to converge with a specific choice of exploring starts. The example is motivated by the counterexample in Bertsekas & Tsitsiklis (1996), where the authors considered estimating the value function in a deterministic continuing-task environment. In Bertsekas & Tsitsiklis (1996), the estimated value function cycles in a fixed region during the iteration with a specific update rule and does not converge. However, that example is not for an episodic task. Our example is concerned with estimating the Q-function in an episodic MDP setting using the MCES algorithm. We can not directly apply the approach in Bertsekas & Tsitsiklis (1996) to our problem without substantial modification. Moreover, in our example, each generated episode and return are stochastic. So it is more difficult to confine the Q-values in a specific region than the deterministic case. These features make our example substantially different from the one in Bertsekas & Tsitsiklis (1996). Next, we formulate the episodic MDP and describe the choice of exploring starts; then we show that the MCES algorithm (4) does not converge with such a choice of exploring starts.

## F.1    MDP FORMULATION

Let the state space be $\mathcal{S} = \{1, 2, 3\}$ where state 3 represents the terminal state, and action space $\mathcal{A} = \{m = move, s = stay\}$. At each state $i$ with $i = 1, 2$, there are two actions, move to the other state or stay. We assume that after taking each action, the system will transit to the terminal state 3 with the same probability $\epsilon > 0$, where $\epsilon$ is a small number. Therefore we have the following transition probability:

$$
\begin{aligned}
p(2|1, m) &= 1 - \epsilon, \quad p(1|1, s) = 1 - \epsilon \\
p(1|2, m) &= 1 - \epsilon, \quad p(2|2, s) = 1 - \epsilon \\
p(3|i, m) &= \epsilon, \quad p(3|i, s) = \epsilon \quad \text{for} \quad i = 1, 2
\end{aligned}
\tag{22}
$$

We also assume the reward function $r(i, a)$ to be:

$$
\begin{aligned}
r(i, m) &= 0 \quad \text{for} \quad i = 1, 2 \\
r(i, s) &= -1 \quad \text{for} \quad i = 1, 2
\end{aligned}
\tag{23}
$$

and the return $G$ is the sum of the reward with the discounted factor $0 < \gamma < 1$.

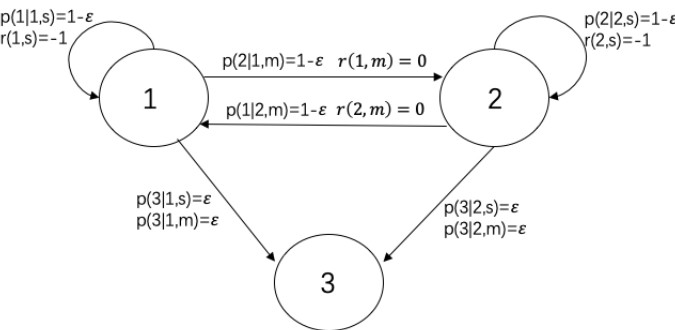

Figure 14: The MDP problem

From the above setting, we can easily see that there are four possible stationary policies:

(a) At each state, always choose to move.

(b) At state 1, move to state 2. At state 2, stay.

(c) At state 1, stay. At state 2, move to state 1.

(d) At each state, always choose to stay.

We denote these four policies by $\pi_a, \pi_b, \pi_c$, and $\pi_d$, respectively. It is easy to see that $\pi_a$ is the optimal policy, and the corresponding action-value function $q_{\pi_a}$ is:

$$
\begin{aligned}
q_{\pi_a}(1, m) = 0, &\quad q_{\pi_a}(1, s) = -1 \\
q_{\pi_a}(2, m) = 0, &\quad q_{\pi_a}(2, s) = -1
\end{aligned}
\tag{24}
$$

### F.2 MECS ALGORITHM WITH A SPECIFIC EXPLORING STARTS

We now apply the MCES algorithm 4 to this MDP problem to estimate the optimal policy $\pi^*(s)$ and the corresponding action-value function $q^*(s, a)$. Note that we have modified the algorithm a little so that for each episode the Q-function is updated only for the initial state-action pair $(S_0, A_0)$.

---

**Algorithm 4** MCES

1: Initialize: $\pi(s) \in \mathcal{A}, Q(s, a) \in \mathbb{R}$, for all $s \in \mathcal{S}, a \in \mathcal{A}$, arbitrarily;
   $Returns(s, a) \leftarrow$ empty list, for all $s \in \mathcal{S}, a \in \mathcal{A}$.
2: **while** True **do**
3:    Choose $S_0 \in \mathcal{S}, A_0 \in \mathcal{A}$ s.t. all pairs are chosen infinitely often.
4:    Generate an episode following $\pi$: $S_0, A_0, S_1, A_1, \ldots, S_{T-1}, A_{T-1}, S_T$.
5:    $G \leftarrow \sum_{t=0}^{T} \gamma^t r(S_t, A_t)$
6:    Append $G$ to $Returns(S_0, A_0)$
7:    $Q(S_0, A_0) \leftarrow average(R(S_0, A_0))$
8:    $\pi(S_0) \leftarrow \arg\max_a Q(S_0, a)$

---

The estimates for $\pi^*(s)$ and $q^*(s, a)$ at the end of $u$th iteration are denoted by $\pi_u(s)$ and $Q_u(s, a)$, respectively. For simplicity, we also denote $Q_u(1, m), Q_u(1, s), Q_u(2, m)$, and $Q_u(2, s)$ by $Q_{1m}^u$, $Q_{1s}^u, Q_{2m}^u$, and $Q_{2s}^u$, respectively. Moreover, the vector $(Q_{im}^u, Q_{is}^u)$ is denoted by

$$
Q_i^u = (Q_{im}^u, Q_{is}^u), \text{ where } i = 1, 2
\tag{25}
$$

The policy $\pi(i)$ is uniquely determined at each state $i$ by comparing the paired value $(Q_{im}^u, Q_{is}^u)$. For example, when $Q_{is} > Q_{im}, \pi(i) = stay$. When $Q_{is} < Q_{im}, \pi(i) = move$. Note that once the policy $\pi$ is determined, the returns of the generated episodes are i.i.d. random variables. There are total four possible cases by comparing paired Q-values $(Q_{1s}, Q_{1m})$ and $(Q_{2s}, Q_{2m})$. The trajectory and return of each case generated by MCES algorithm 4 are listed in Table 1 and 2.

| State 1 | State 2 | Initial start | Episode trajectory | Return |
|---|---|---|---|---|
| Q(1,s)>Q(1,m) | Q(2,s)>Q(2,m) | (1,s) | 1,s,1,s,1,s... | $-1-\gamma-\gamma^2$... |
| | | (1,m) | 1,m,2,s,2,s... | $0-\gamma-\gamma^2$... |
| | | (2,s) | 2,s,2,s,2,s... | $-1-\gamma-\gamma^2$... |
| | | (2,m) | 2,m,1,s,1,s... | $0-\gamma-\gamma^2$... |
| Q(1,s)>Q(1,m) | Q(2,s)<Q(2,m) | (1,s) | 1,s,1,s,1,s... | $-1-\gamma-\gamma^2$... |
| | | (1,m) | 1,m,2,m,1,s,1,s... | $0-0-\gamma^2-\gamma^3$... |
| | | (2,s) | 2,s,2,m,1,s,1,s... | $-1-0-\gamma^2-\gamma^3$... |
| | | (2,m) | 2,m,1,s,1,s... | $0-\gamma-\gamma^2$... |

Table 1: Episode and return with different initial starts generated by the MCES algorithm when $Q(1, s) > Q(1, m)$

| State 1 | State 2 | Initial start | Episode trajectory | Return |
|---|---|---|---|---|
| Q(1,s)<Q(1,m) | Q(2,s)>Q(2,m) | (1,s) | 1,s,1,m,2,s,2,s... | $-1-0-\gamma^2-\gamma^3...$ |
| | | (1,m) | 1,m,2,s,2,s... | $0-\gamma-\gamma^2...$ |
| | | (2,s) | 2,s,2,s,2,s... | $-1-\gamma-\gamma^2...$ |
| | | (2,m) | 2,m,1,m,2,s,2,s... | $0-0-\gamma^2-\gamma^3...$ |
| Q(1,s)<Q(1,m) | Q(2,s)<Q(2,m) | (1,s) | 1,s,1,m,2,m,1,m... | -1 |
| | | (1,m) | 1,m,2,m,1,m,1,m... | 0 |
| | | (2,s) | 2,s,2,m,1,m,2,m... | -1 |
| | | (2,m) | 2,m,1,m,2,m... | 0 |

Table 2: Episode and return with different initial starts generated by the MCES algorithm when $Q(1, s) < Q(1, m)$

Here we give the expected returns of several cases related to our counterexample.

**Lemma 2.** *Consider an episode generated by the MCES algorithm 4:*

*(i) When $Q(1, s) < Q(1, m)$, $Q(2, s) > Q(2, m)$, and the initial start is $(1, m)$, or when $Q(1, s) > Q(1, m)$, $Q(2, s) > Q(2, m)$, and the initial start is $(2, m)$, the expected returns $q_{\pi_b}(1, m)$ and $q_{\pi_d}(2, m)$ are the same and are equal to*

$$-\mu_1 := \mathbb{E}[G] = \frac{-\gamma(1 - \epsilon)}{1 - \gamma + \gamma\epsilon} \tag{26}$$

*(ii) When $Q(1, s) > Q(1, m)$, $Q(2, s) < Q(2, m)$, and initial start is $(2, s)$, or when $Q(1, s) < Q(1, m)$, $Q(2, s) > Q(2, m)$, and the initial start is $(1, s)$, the expected returns $q_{\pi_c}(2, s)$ and $q_{\pi_b}(1, s)$ are the same and are equal to*

$$-\mu_2 := \mathbb{E}[G] = \frac{-1}{1 - \gamma + \gamma\epsilon} + \gamma(1 - \epsilon) \tag{27}$$

*(iii) When $Q(1, s) > Q(1, m)$ and the initial start is $(1, s)$, or when $Q(2, s) > Q(2, m)$ and the initial start is $(2, s)$, the expected return is*

$$-\mu_3 := \mathbb{E}[G] = \frac{-1}{1 - \gamma + \gamma\epsilon} \tag{28}$$

*Moreover, we have $\mu_1 < \mu_2 < \mu_3$, and the variance of the return in each case is finite.*

*Proof.* Let $A_N$ be the event that a generated episode terminates after taking $N + 1$ actions. From the formulation of our MDP problem, we have

$$P(A_N) = \epsilon(1 - \epsilon)^N, \ N \geq 0 \tag{29}$$

We first prove (i), in this case, the episode is of the form:

$$1, m, 2, s, 2, s, ... \text{ or } 2, m, 1, s, 1, s, ...$$

the return of an episode that terminates after taking $N + 1$ actions is

$$G = -\frac{\gamma(1 - \gamma^N)}{1 - \gamma}, \ N \geq 0 \tag{30}$$

so

$$\mathbb{E}[G] = \sum_{N=0}^{\infty} -\frac{\gamma(1 - \gamma^N)}{1 - \gamma}\epsilon(1 - \epsilon)^N = -\frac{\gamma}{1 - \gamma} + \frac{\epsilon\gamma}{(1 - \gamma)(1 - \gamma + \gamma\epsilon)} = \frac{-\gamma(1 - \epsilon)}{1 - \gamma + \gamma\epsilon} \tag{31}$$

(ii) In this case the episode is of the form

$$1, s, 1, m, 2, s, 2, s, \dots \text{ or } 2, s, 2, m, 1, s, 1, s \dots$$

the probability density of the return is

$$P(G = -1) = P(A_0 \cup A_1) = \epsilon + \epsilon(1 - \epsilon)$$

$$P\left(G = -1 - \frac{\gamma^2 - \gamma^{N+1}}{1 - \gamma}\right) = P(A_N) = \epsilon(1 - \epsilon)^N, \text{ for } N \geq 2 \tag{32}$$

the expected return is

$$
\begin{aligned}
\mathbb{E}[G] &= -\epsilon - \epsilon(1 - \epsilon) + \sum_{N=2}^{\infty} \left(-1 - \frac{\gamma^2 - \gamma^{N+1}}{1 - \gamma}\right)\epsilon(1 - \epsilon)^N \\
&= -\sum_{N=0}^{\infty} \frac{1 - \gamma^{N+1}}{1 - \gamma}\epsilon(1 - \epsilon)^N + \gamma\epsilon \sum_{N=1}^{\infty} (1 - \epsilon)^N \\
&= -\frac{1}{1 - \gamma} + \frac{\epsilon\gamma}{(1 - \gamma)(1 - \gamma + \gamma\epsilon)} + \gamma(1 - \epsilon) \\
&= \frac{-1}{1 - \gamma + \gamma\epsilon} + \gamma(1 - \epsilon)
\end{aligned}
\tag{33}
$$

(iii) In this case the possible episode is

$$1, s, 1, s, 1, s, \dots \text{ or } 2, s, 2, s, 2, s, \dots$$

so the probability density of the return is

$$P\left(G = -\frac{1 - \gamma^{N+1}}{1 - \gamma}\right) = P(A_N) = \epsilon(1 - \epsilon)^N, \ N \geq 0 \tag{34}$$

and the expectation is

$$
\begin{aligned}
\mathbb{E}[G] &= \sum_{N=0}^{\infty} -\frac{1 - \gamma^{N+1}}{1 - \gamma}\epsilon(1 - \epsilon)^N \\
&= -\frac{1}{1 - \gamma} + \frac{\epsilon\gamma}{(1 - \gamma)(1 - \gamma + \gamma\epsilon)} \\
&= \frac{-1}{1 - \gamma + \gamma\epsilon}
\end{aligned}
\tag{35}
$$

Moreover, we have

$$\frac{\gamma(1 - \epsilon)}{1 - \gamma + \gamma\epsilon} < \frac{1}{1 - \gamma + \gamma\epsilon} - \gamma(1 - \epsilon) < \frac{1}{1 - \gamma + \gamma\epsilon} \tag{36}$$

By a similar calculation, we can see that the second moment $\mathbb{E}[G^2]$ of each case is finite. Therefore, the variances are also finite. □

Next, we describe specific regions in $Q_{1m}$-$Q_{1s}$ and $Q_{2m}$-$Q_{2s}$ planes. The two planes and the corresponding regions are represented in Fig. 15a and Fig. 15b, respectively. In each figure, $Q_{im}$-axis and $Q_{is}$-axis represent the value of $Q_{im}^u$ and $Q_{is}^u$ after $u$th iteration given the state $i$, where $i = 1, 2$. Since $\pi(i) = \arg\max_a Q(i, a)$, we can see that the line $l_i$

$$l_i : Q_{im} = Q_{is} \tag{37}$$

is the boundary for different policies $\pi(i)$. For the vector $(Q_{im}, Q_{is})$ above the line $l_i$, we have $\pi(i) = stay$. Similarly, for those below $l_i$, we have $\pi(i) = move$. In the $Q_{1m}$-$Q_{1s}$ plane, we define

$$\mathcal{R}_1 = \{(Q_{1m}^u, Q_{1s}^u) : Q_{1m}^u > Q_{1s}^u, Q_{1s}^u < -\mu_2 + \delta, Q_{1m}^u < -b_1\} \tag{38}$$

$$\mathcal{R}_2 = \{(Q_{1m}^u, Q_{1s}^u) : Q_{1s}^u < -\mu_2 + \delta, -b_1 < Q_{1m}^u < -1\} \tag{39}$$

$$\mathcal{R}_3 = \{(Q_{1m}^u, Q_{1s}^u) : Q_{1m}^u < Q_{1s}^u, -b_1 < Q_{1m}^u < -1, Q_{1s}^u < -1\} \tag{40}$$

and in the $Q_{2m}$-$Q_{2s}$ plane, we define

$$\mathcal{T}_1 = \{(Q_{2m}^u, Q_{2s}^u) : -1 < Q_{2m}^u < 0, Q_{2s}^u < -\mu_1 + \delta\} \tag{41}$$

$$\mathcal{T}_2 = \{(Q_{2m}^u, Q_{2s}^u) : -1 < Q_{2m}^u < 0, -1 - \delta < Q_{2s}^u < -1\} \tag{42}$$

$$\mathcal{T}_3 = \{(Q_{2m}^u, Q_{2s}^u) : -\mu_1 - \delta < Q_{2m}^u < -\mu_1 + \delta, Q_{2s}^u > Q_{2m}^u, Q_{2s}^u < -1\} \tag{43}$$

$$\mathcal{T}_4 = \{(Q_{2m}^u, Q_{2s}^u) : -\mu_1 - \delta < Q_{2m}^u < -\mu_1 + \delta, Q_{2s}^u < Q_{2m}^u\} \tag{44}$$

Here the parameters $\delta$ and $b_1$ satisfy

$$0 < \delta < \min\{\frac{\mu_1 - 1}{2}, \frac{\mu_2 - \mu_1}{2}, \mu_2 - 1 - \frac{\gamma}{2(1-\gamma)}\}, \ 2 < b_1 < \min\{\mu_1, 2\mu_2 - 2\delta - \frac{\gamma}{1-\gamma}\} \tag{45}$$

The regions $\mathcal{R}_i$ and $\mathcal{T}_i$ are nonempty if $\gamma$ and $\epsilon$ satisfy the following condition:

**Condition 3.**

$$\frac{\gamma(1-\epsilon)}{1 - \gamma + \gamma\epsilon} > 2 \tag{46}$$

*and*

$$\frac{1}{1 - \gamma + \gamma\epsilon} - \frac{\gamma}{2(1-\gamma)} - \gamma(1-\epsilon) - 1 > 0 \tag{47}$$

**Remark 4.** *We claim that there exist $\gamma$ and $\epsilon$ that satisfy Condition 3. Here we give one way to find suitable $\gamma$ and $\epsilon$. Let the ratio $\frac{1-\gamma}{\epsilon} = 10$, then $\epsilon = \frac{1-\gamma}{10}$ and*

$$\frac{\gamma(1-\epsilon)}{1 - \gamma + \gamma\epsilon} = \frac{\gamma(9+\gamma)}{10 - \gamma(9+\gamma)}$$

*as $\gamma$ approaches 1, $\frac{\gamma(9+\gamma)}{10-\gamma(9+\gamma)}$ approaches infinity. So we can choose $\gamma$ close to 1 (then $\epsilon$ is close to 0) such that $\frac{\gamma(1-\epsilon)}{1-\gamma+\gamma\epsilon} > 2$. Particularly, we can choose $0.7 < \gamma < 1$ such that (46) holds. For the second inequality, similarly*

$$\frac{1}{1 - \gamma + \gamma\epsilon} - \frac{\gamma}{2(1-\gamma)} = \frac{\gamma^2 + 10\gamma - 20}{2(\gamma^2 + 9\gamma - 10)}$$

*it approaches to infinity as $\gamma$ approaches 1. So we see that Condition 3 will hold for all $0.7 < \gamma < 1$ when $\epsilon = (1 - \gamma)/10$.*

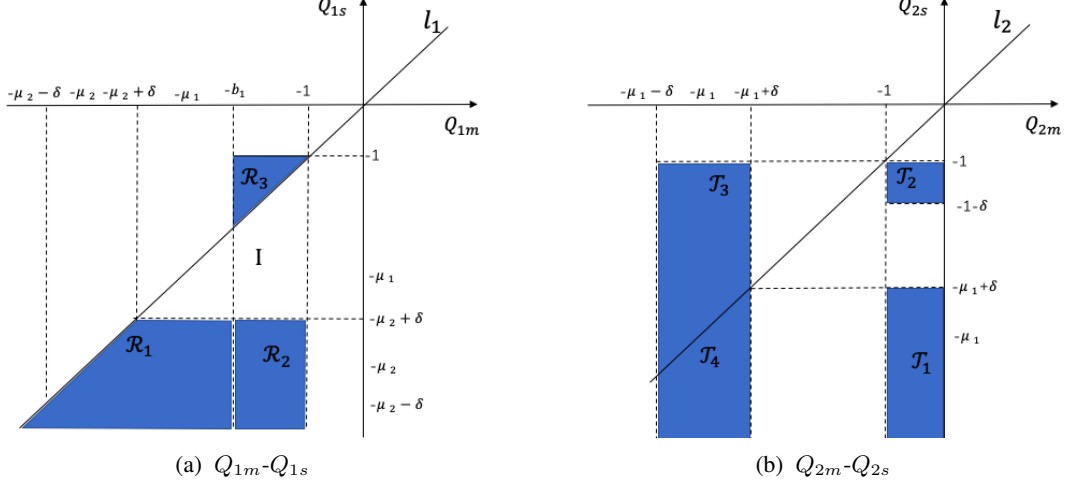

(a) $Q_{1m}$-$Q_{1s}$        (b) $Q_{2m}$-$Q_{2s}$

Figure 15: $\mathcal{R}_i$ and $\mathcal{T}_i$ are blue regions in $Q_{1m}$-$Q_{1s}$ plane and $Q_{2m}$-$Q_{2s}$ plane

Recall that $Q_1^u = (Q_{1m}^u, Q_{1s}^u)$ and $Q_2^u = (Q_{2m}^u, Q_{2s}^u)$. Assuming that $Q_1^u$ and $Q_2^u$ are initialized in $\mathcal{R}_1$ and $\mathcal{T}_1$, respectively, we now describe the specific choice of exploring starts.

**Exploring starts 5.** *We initialize $Q_1^u$ and $Q_2^u$ to be in $\mathcal{R}_1$ and $\mathcal{T}_1$, respectively. We then repeat the following 8 steps over and over again:*

1. *Keep choosing $(1, m)$ as the initial state-action pair and update $Q(1, m)$ until $Q_1^u$ enters $\mathcal{R}_2$.*

2. *Keep choosing $(2, s)$ as the initial state-action pair and update $Q(2, s)$ until $Q_2^u$ enters $\mathcal{T}_2$.*

3. *Keep choosing $(1, s)$ as the initial state-action pair and update $Q(1, s)$ until $Q_1^u$ enters $\mathcal{R}_3$.*

4. *Keep choosing $(2, m)$ as the initial state-action pair and update $Q(2, m)$ until $Q_2^u$ enters $\mathcal{T}_3$.*

5. *Keep choosing $(1, s)$ as the initial state-action pair and update $Q(1, s)$ until $Q_1^u$ enters $\mathcal{R}_2$.*

6. *Keep choosing $(1, m)$ as the initial state-action pair and update $Q(1, m)$ until $Q_1^u$ returns to $\mathcal{R}_1$.*

7. *Keep choosing $(2, s)$ as the initial state-action pair and update $Q(2, s)$ until $Q_2^u$ enters $\mathcal{T}_4$.*

8. *Keep choosing $(2, m)$ as the initial state-action pair and update $Q(2, m)$ until $Q_2^u$ returns to $\mathcal{T}_1$.*

*Note that in each of 8 steps, only one of the values $Q(1, m)$, $Q(1, s)$, $Q(2, m)$, and $Q(2, s)$ changes. All others remain constant.*

The trajectories of $Q_1^u$ and $Q_2^u$ during the iteration are visualized in Fig. 16a and Fig. 16b, respectively.

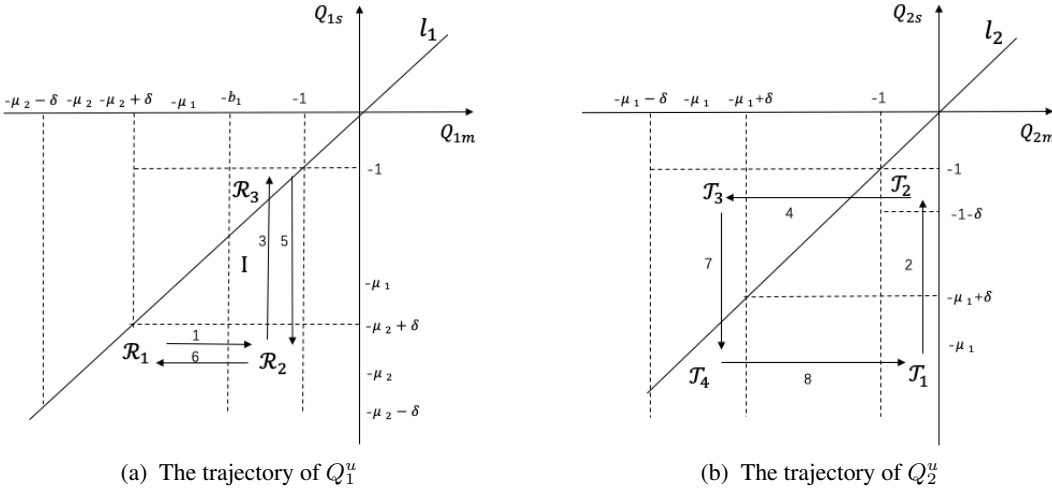

(a) The trajectory of $Q_1^u$          (b) The trajectory of $Q_2^u$

Figure 16: Trajectories of Q-values during the iteration

Next, we show that the MCES algorithm 4 does not converge with the exploring starts (5). Precisely, we have the following result

**Theorem 6.** *Given $\gamma$ and $\epsilon$ satisfy the condition (3), suppose that $Q_1^u$ and $Q_2^u$ are initialized in $\mathcal{R}_1$ and $\mathcal{T}_1$, respectively. Following the MCES algorithm (4) with the exploring starts 5, the trajectory of $Q_1^u$ forms a cycle according to $\mathcal{R}_1 \to \mathcal{R}_2 \to \mathcal{R}_3 \to \mathcal{R}_2 \to \mathcal{R}_1$ almost surely, and the trajectory of $Q_2^u$ forms a cycle according to $\mathcal{T}_1 \to \mathcal{T}_2 \to \mathcal{T}_3 \to \mathcal{T}_4 \to \mathcal{T}_1$ almost surely. Thus, $Q_u(s, a)$ does not converge with probability 1.*

*Proof.* **Step 1. Start with $(1, m)$**

When $Q_1^u \in \mathcal{R}_1$ and $Q_2^u \in \mathcal{T}_1$, we have $Q(1, s) < Q(1, m)$ and $Q(2, s) < Q(2, m)$, the generated episode is just

$$1, m, 2, m, 1, m...$$

so the return is always 0. During the iteration, $Q_{1m}^u$ increases and approaches zero according to

$$Q_{1m}^{u+1} = (1 - \frac{1}{n})Q_{1m}^u + \frac{1}{n} \cdot 0 \tag{48}$$

Here $n$ denote the number of stored $Q_{1m}$-values after the $(u+1)$-th iteration. Since when $Q_1^u \in \mathcal{R}_1$, $Q_{1m}^u < -b_1 < -2$, and $n \geq 2$, we have

$$Q_{1m}^{u+1} = (1 - \frac{1}{n})Q_{1m}^u < (1 - \frac{1}{n})(-2) < -1 \tag{49}$$

So when $Q_1^u \in \mathcal{R}_1$, we have $Q_{1m}^{u+1} < -1$, and $Q_1^u$ will stay in $\mathcal{R}_1$ or $\mathcal{R}_2$ during the iteration. Since $Q_{1m}^u$ approaches zero, after several updates, $Q_1^u$ enters the region $\mathcal{R}_2$.

**Step 2. Start with** $(2, s)$

When $Q_1^u \in \mathcal{R}_2$ and $Q_2^u \in \mathcal{T}_1$, the return with initial start $(2, s)$ is always $-1$. So during the iteration, $Q_{2s}^u$ will increase and approach $-1$. Eventually, $Q_2^u$ will enter $\mathcal{T}_2$.

**Step 3. Start with** $(1, s)$

Similar to step 2, the return is always $-1$, and $Q_{1s}^u$ will approach $-1$. Since in $\mathcal{R}_2$ we have $Q_{1m}^u < -1$, $Q_1^u$ will cross the line $l_1$ and enter $\mathcal{R}_3$ after several updates.

**Step 4. Start with** $(2, m)$

In this case, $Q_1^u \in \mathcal{R}_3$ and $Q_2^u \in \mathcal{T}_2$, so every episode is generated by the same policy and is of the form:

$$2, m, 1, s, 1, s, ...$$

So the returns are i.i.d. random variables, and from lemma (2), we have

$$\mathbb{E}[G] = \frac{-\gamma(1 - \epsilon)}{1 - \gamma + \gamma\epsilon} = -\mu_1 \tag{50}$$

Let $n$ be the number of episodes generated in this step and $G_i$ be their returns, then

$$S_n = G_1 + G_2 + ... + G_n$$

By the strong law of large numbers, we have

$$\frac{S_n}{n} \to -\mu_1, \text{ a.s. } (\text{as } n \to +\infty) \tag{51}$$

Assume that we have had $t$ iterations before we start step 4, and there are $L$ stored $Return(2, m)$, then

$$Q_{2m}^{t+n} = \frac{LQ_{2m}^t + S_n}{L + n} = \frac{LQ_{2m}^t}{L + n} + \frac{S_n}{L + n} \tag{52}$$

since $L$ and $Q_{2m}^t$ are finite numbers and independent of $n$, we have

$$\lim_{n \to \infty} Q_{2m}^{t+n} = -\mu_1 \text{ a.s.} \tag{53}$$

Therefore, when we keep choosing $(2, m)$ as the initial start and updating $Q(2, m)$, we will have $Q_{2m}^{t+n} < -\mu_1 + \delta$ for some $n$. Thus, $Q_2^u$ will enter the region $\mathcal{T}_3$ almost surely after several iterations. Note that even though the policy $\pi_u(2)$ changes after $Q_2^u$ crosses the line $l_2$, it does not affect the return of the generated episode with the initial state-action pair $(2, m)$ since only the first state of the generated episode is state 2.

**Step 5. Start with** $(1, s)$

In this case, at first we have

$$Q_{1s}^u > Q_{1m}^u \text{ and } Q_{2s}^u > Q_{2m}^u \tag{54}$$

therefore $\pi_u(1) = stay$ and $\pi_u(2) = stay$. The generated episode with initial $(1, s)$ is of the form:

$$1, s, 1, s, 1, s, ... \tag{55}$$

and the returns are i.i.d. random variables with the expectation:

$$\mathbb{E}[G] = \frac{-1}{1 - \gamma + \gamma\epsilon} = -\mu_3 \tag{56}$$

Similar to the argument in step 4, when we keep updating $Q_{1s}^u$ by choosing $(1, s)$ as the initial start, $Q_{1s}^u$ can not stay in $\mathcal{R}_3$ forever. It will approach $-\mu_3$. So $Q_1^u$ will cross the line $l_1$ a.s.

After crossing the line $l_1$, $Q_1^u$ will enter $\mathcal{R}_2$ or $\mathcal{I}$, where $\mathcal{I}$ is the region

$$\mathcal{I} = \{(Q_{1m}^u, Q_{1s}^u) : -b_1 < Q_{1m}^u < -1, Q_{1m}^u > Q_{1s}^u, Q_{1s}^u > -\mu_2 + \delta\} \tag{57}$$

If $Q_1^u$ enters $\mathcal{R}_2$, we end this step and move to the next. Otherwise, we still keep updating $Q_{1s}^u$. At this moment, we have

$$Q_{1s}^u < Q_{1m}^u \text{ and } Q_{2s}^u > Q_{2m}^u \tag{58}$$

the generated episode then becomes

$$1, s, 1, m, 2, s, 2, s, 2, s... \tag{59}$$

and the returns are i.i.d. random variables from another distribution with the expectation:

$$\mathbb{E}[G] = \frac{-1}{1 - \gamma + \gamma\epsilon} + \gamma(1 - \epsilon) = -\mu_2 \tag{60}$$

so if $Q_1^u$ stays in $\mathcal{I}$ during the iteration, $Q_{1s}^u$ will approach $-\mu_2$. On the other hand, since we can successively get returns equal to -1, $Q_1^u$ might return to $\mathcal{R}_3$. Then the generated episodes are of type (55), $Q_1^u$ will again cross the line $l_1$ after several updates. We claim that $Q_1^u$ can not oscillate between $\mathcal{R}_3$ and $\mathcal{I}$ forever. If so, we will get infinitely many sample returns with the expectation either $-\mu_2$ or $-\mu_3$, the sample mean, $Q_{1s}^u$ will be close to a convex combination of $-\mu_2$ and $-\mu_3$. And we will have $Q_{1s}^u < -\mu_2 + \delta$ for some $u$. Therefore, $Q_1^u$ enters $\mathcal{R}_2$ after several updates eventually.

**Step 6. Start with $(1, m)$**

The possible episode in this case is

$$1, m, 2, s, 2, s, 2, s...$$

therefore the expectation of the return is $-\mu_1$. During the iteration, $Q_{1m}^u$ will converge to $-\mu_1$. So $Q_{1m}^u$ will decrease, and $Q_1^u$ will return to $\mathcal{R}_1$. Note that the sample return satisfies

$$G > -\frac{\gamma}{1 - \gamma} \tag{61}$$

When $Q_1^u \in \mathcal{R}_2$, the value of $Q_{1m}^{u+1}$ is at least $-\frac{1}{2}(b_1 + \frac{\gamma}{1-\gamma})$. By condition (3) and our assumptions (45) on $b_1$ and $\delta$, we have

$$Q_{1m}^{u+1} > -\mu_2 + \delta \tag{62}$$

Therefore, when $Q_1^u \in \mathcal{R}_2$, $Q_1^{u+1}$ will not cross the line $l_1$ during the iteration, and the policy $\pi_u$ does not change. This ensures that we can get returns from the same distribution so that $Q_1^u$ will return to $\mathcal{R}_1$.

**Step 7. Start with $(2, s)$**

In this case, we have $Q_1^u \in \mathcal{R}_1$, and $Q_2^u \in \mathcal{T}_3$, so the possible episode is:

$$2, s, 2, s, 2, s...$$

The expected return is $-\mu_3$. Similar to the previous discussion, after several updates, $Q_2^u$ will cross the line $l_2$ and enter $\mathcal{T}_4$, eventually.

**Step 8. Start with $(2, m)$**

When $Q_1^u \in \mathcal{R}_1$ and $Q_2^u \in \mathcal{T}_4$, the return of an episode starts with $(2, m)$ is always 0. So when we keep updating $Q_{2m}^u$, it will approach 0, and $Q_2^u$ will return to $\mathcal{T}_1$.

After $Q_2^u$ returns to $\mathcal{T}_1$, we go back to step 1 and follow the eight-step exploring starts (5) again. We can see that during this process, all the state-action pairs can be chosen infinitely often, and the Q-values will continue to alternate between these regions and not converge. Also, the policy does not converge. Therefore, Algorithm (4) following the exploring starts (5) does not converge with probability 1. □

