# OpenReview forum: "On the Convergence of the Monte Carlo Exploring Starts Algorithm for Reinforcement Learning"
_ICLR.cc/2022/Conference — ICLR 2022 Poster_

### Official Review · Reviewer_Eew5 · 2021-10-30

**Correctness:** 3
**Technical Novelty And Significance:** 4
**Empirical Novelty And Significance:** 2
**Recommendation:** 8
**Confidence:** 3

**Main Review:**

Strengths:
    Presentation is mostly clear
    A clear comparison to related works
    An important theoretical contribution to the field.
    Simple idea and proof.
    Empirical results echo the claims to some extend.

Weakness:
   The assumption of the uniqueness of the policy is not presented in the theorem.
   More discussion about the empirical results would be helpful. For example, Tsitsiklis's algorithm requires a uniformly sampling of the start state in order to guarantee convergence, but empirically it also works well with a different way to sample the start state. I would like to see how the authors think about it.
   Some expressions can be improved:  the word "iteration" is not in the algorithm but is in the proof to refer to something in the algorithm.
   Some typos: 1. Algorithm3 line 9, it should be Q_t(S_t, t, A_t), 2. when citing a paper by two authors, sometimes it lists two authors' names and sometimes it uses et al.


**Summary Of The Paper:**

This paper proposed a proof of the Monte Carlo Exploring Starts algorithm given by Sutton and Barto (1998) for a class of MDPs.

**Summary Of The Review:**

Overall I like this paper because of the strengths listed in the main review. There is some weakness but it doesn't weaken the main contribution of the paper.

---

> ### Author Response · Authors · 2021-11-16
> **Response to Reviewer Eew5**
>
> Thank you for your positive evaluation and helpful feedback! Here are our response to your comments:
>
> > The assumption of the uniqueness of the policy is not presented in the theorem.
>
> Thank you for the suggestions. It is not necessary to assume that the optimal policy is unique. We only did so because it streamlines the proof a little.  We will make it more clear in our revision.
>
> >More discussion about the empirical results would be helpful. For example, Tsitsiklis's algorithm requires a uniformly sampling of the start state in order to guarantee convergence, but empirically it also works well with a different way to sample the start state. I would like to see how the authors think about it.
>
> From the experimental results we see that Tsitsiklis's algorithm (only updates the initial (s,a) pair) works in Blackjack even when we sample according to the standard rule of Blackjack and not uniformly. We believe this is because although the standard rule of Blackjack does not sample uniformly from the state space, the initial state distribution under this rule still sufficiently covers the state space of Blackjack, so learning is still possible. However, this is not the case for CliffWalking, where if we initialize an episode according to the standard rule, the agent will always initialize at the bottom left corner of the gridworld, making it impossible to update the value function for other states. We will add more discussion for the empirical results.
>
> >Some expressions can be improved: the word "iteration" is not in the algorithm but is in the proof to refer to something in the algorithm. Some typos: 1. Algorithm3 line 9, it should be Q_t(S_t, t, A_t), 2. when citing a paper by two authors, sometimes it lists two authors' names and sometimes it uses et al.
>
> Thank you for pointing out these minor issues, we will go through the paper carefully and fix them in our revision.

---

> > ### Comment · Reviewer_Eew5 · 2021-11-28
> > **Thanks for you reply.**
> >
> > After reading the author's response, I would like to keep my score.

---

### Official Review · Reviewer_joaK · 2021-10-30

**Correctness:** 4
**Technical Novelty And Significance:** 2
**Empirical Novelty And Significance:** 2
**Recommendation:** 5
**Confidence:** 4

**Main Review:**

Pros:
- The paper provides a novel convergence proof for MCES that relaxes the assumptions that have been previously employed for proving convergence of MCES
- The proof, as the authors highlight, is simple and does not make use of hard mathematical/statistical tools

Cons:
- The classes of MDP considered seem very constraining. Basically, it is required that (in the less restrictive case under the optimal policy) the states are never re-visited along one trajectory. The authors succeed in providing justifications for these assumptions, especially providing several realistic examples in which the assumptions are fulfilled. Nevertheless, I think that these assumptions are ensuring that within one episode a single (s,a)-pair will be updated at most once, significantly simplifying the analysis. My main concern is about the significance of these convergence results under such strong assumptions. To put it another way, is this analysis a step towards the understanding of MCES for general MDPs or just the analysis of a particular restricted case?

Minor issues:
- Pag 4: "Maximizing (2) corresponds to the stochastic shortest path problem". Is it true? I think that this holds only when the reward penalizes the steps needed to reach the goal state, not for general rewards.
- Lemma 1: I think that some condition on the almost sure boundedness of the involved random variables is needed.
- The ticks on the plot axis are too small
- Algorithm 3, line 9: Q should have an additional argument t
- Pag 1:  doesn’t -> does not
- Pag 4:  isn’t -> is not

**Summary Of The Paper:**

The paper studies the convergence of Monte Carlo Exploring Starts (MCES), in which the Q-function is estimated by averaging Monte Carlo returns and the policy is defined as the greedy policy w.r.t. this Q-function. The authors provide a technically simple proof of the convergence under different assumptions on the underlying MDP: (i) stochastic feed-forward MDP (in which states cannot be re-visited in an episode); (ii) optimal policy feed-forward (in which under the optimal policy states are not re-visited); (iii) finite-horizon MDPs. Given these additional assumptions, the analysis relaxes other assumptions employed in previous convergence proofs. Finally, experimental results to validate the considered set of assumptions is provided.

**Summary Of The Review:**

Overall, I think that the paper provides a novel contribution. However, given that the considered assumptions are, in my opinion, very restrictive, I have concerns about the significance of the theoretical results. For these reasons, I opt for a borderline score.

---

> ### Author Response · Authors · 2021-11-16
> **Response to Reviewer joaK**
>
> Thank you for your very detailed comments and questions! Here are our response to your concerns:
>
> >The classes of MDP considered seem very constraining. Basically, it is required that (in the less restrictive case under the optimal policy) the states are never re-visited along one trajectory. The authors succeed in providing justifications for these assumptions, especially providing several realistic examples in which the assumptions are fulfilled. Nevertheless, I think that these assumptions are ensuring that within one episode a single (s,a)-pair will be updated at most once, significantly simplifying the analysis. My main concern is about the significance of these convergence results under such strong assumptions. To put it another way, is this analysis a step towards the understanding of MCES for general MDPs or just the analysis of a particular restricted case?
>
> Thank you for your comments and this is a very good question. We would like to first point out that for optimal policy feed-forward (OPFF) MDPs (the class of MDP that we are mainly concerned with), if the current policy is not the optimal policy, then within one episode, a single (s,a)-pair can in fact be updated an **arbitrary number of times** and not just **once**. This is because the OPFF assumption does not put constraints on sub-optimal actions; when the agent takes a sub-optimal action, it can transition into an arbitrary state in the MDP.
>
> It is correct that a single (s,a)-pair can be updated at most once when **under the optimal policy**. However, we never assumed that, when the MCES started learning, we already converged to the optimal policy (if that is the case, the learning problem would be immediately solved, and we don't really need to run MCES, or any other learning algorithm). We would like to emphasize that when the learning starts, the agent initializes with a **random policy,** and will be taking sub-optimal actions all the time, leading to an arbitrary number of updates for each (s-a)-pair in the episode, with arbitrary return values. So for OPFF MDPs, we in fact **do not** have this constraint.
>
> We would like to further point out that, even when **under the constraint** that "a single (s,a)-pair can be updated at most once," proving the convergence of the MCES algorithm is **still not trivial**. In fact, even under the **much stronger assumption** that for each episode, only the first (s-a) pair in the episode is updated just once (first-visit MCES), the convergence for the MCES algorithm is **still not guaranteed**. Indeed, in Chapter 5 of [1], a counterexample is given to show that in a simple 2-state MDP (which is not OPFF), MCES can fail to converge to the optimal policy, even under the stronger assumption that for each episode, only the first (s-a) pair is updated once.
>
> To summarize,
> 1) OPFF MDPs in fact do not have this constraint;
> 2) even when this constraint is applied, or even made stronger, convergence is still not guaranteed;
> 3) as discussed in the paper, many realistic problems fall into the category of OPFF MDPs.
>
> Based on the above facts, we believe our convergence results are not trivial and indeed make an important step towards understanding MCES for general MDPs.
>
> > Minor issues:
> > Pag 4: "Maximizing (2) corresponds to the stochastic shortest path problem". Is it true? I think that this holds only when the reward penalizes the steps needed to reach the goal state, not for general rewards.
> Lemma 1: I think that some condition on the almost sure boundedness of the involved random variables is needed.
> The ticks on the plot axis are too small
> Algorithm 3, line 9: Q should have an additional argument t
> Pag 1: doesn’t -> does not
> Pag 4: isn’t -> is not
>
> Thank you for pointing out these minor issues.
>
> The stochastic shortest path (SSP) problem has been defined in [1] and [2] with general rewards and we are fully consistent with their definition. For the other minor issues, we will go over the paper carefully and address them in our revision.
>
> [1] Bertsekas and Tsitsiklis. Neuro-dynamic programming, volume 5. Athena Scientific Belmont, MA, 1996.
>
> [2] Bertsekas and Tsitsiklis. An analysis of stochastic shortest path problems. Mathematics of Operations Research, 16(3):580–595, 1991.

---

> > ### Comment · Reviewer_joaK · 2021-11-27
> > **Re: Response to Reviewer joaK**
> >
> > I thank the authors for the answer and, in particular, for having clarified the peculiarities of the setting. Nevertheless, I am still unsure whether the proposed analysis is really "an important step towards understanding MCES for general MDPs". I currently fear that the enforced conditions (e.g., OPFF MDP) are fundamental for the analysis. My feeling is that relaxing these assumptions would invalidate the basic principles behind the current analysis. If I understood well, the basic idea of the proof of the theorems is a kind of "progressive convergence" in which states that are visited later under the optimal policy are characterized by a Q-function that converges sooner. I think this idea cannot be exploited for general MDPs, and it is not easily adaptable. Nevertheless, I recognize that the paper provides the first convergence proof for the MCES algorithm, though for a quite constrained class of MPDs.

---

> > > ### Author Response · Authors · 2021-11-28
> > > **Updated response to Reviewer joaK**
> > >
> > > Thank you for your response and interesting comments. We agree that the OPFF assumption is fundamental to the analysis, and the proof technique presented in this paper very much relies on this assumption. However, we would like to emphasize the following important points:
> > >
> > > 1. As discussed in the paper, there are many natural examples of OPFF MDPs, including those used by benchmarks in other research papers (Blackjack, MuJoCo robotic locomotion, windy gridworlds as described in our appendix). In fact, we believe that most episodic MDPs of practical interest are OPFF. Also note that if a task involves any monotonically changing value as part of the state, then it is also OPFF. For example, when time is added to the state to handle finite horizon criteria, then the MDP becomes OPFF whether or not the original MDP is OPFF.
> > >
> > >
> > >     We can in fact give an extended list of real-world practical problems that fall into OPFF MDPs:
> > >     - Operating a robot or datacenter with a power budget;
> > >     - Driving a car with a given amount of fuel or to reach a target within a time limit;
> > >     - Manufacturing a product with limited resources;
> > >     - Doing online ads bidding with a fixed budget;
> > >     - Running a recommendation systems with a limited amount of recommendation attempts;
> > >     - Trading to maximize profit within a time period, and more;
> > >
> > >     All these tasks fall into the category of OPFF MDPs. Therefore the class of OPFF MDPs is a large and important one.
> > >
> > > 2. As discussed in the Introduction of our paper, it is known the original MCES algorithm does not converge for some MDPs outside the class of OPFF. It may be that for the original MCES algorithm, non-OPFF MDPs do not converge in general, or at least a large class of non-MDPs do not converge. Although this has not been established, it may be the case that no proof technique will be able to establish convergence of non-OPFF MDPs. This is an area for future research.
> > >
> > > 3. We would like to invite you to carefully read the proof of convergence of OPFF MDPs (in the Appendix.) We believe you will find the proof to be elegant but not entirely straightforward since it needs to overcome several subtle and challenging obstacles (see our earlier rebuttal). Also, unlike the proofs on dynamic programming, it is an almost-sure convergence proof, and has a  fresh style and flavor.

---

### Official Review · Reviewer_U5JN · 2021-10-31

**Correctness:** 3
**Technical Novelty And Significance:** 2
**Empirical Novelty And Significance:** 2
**Recommendation:** 5
**Confidence:** 4

**Main Review:**

The paper is written well; I enjoyed how it provides a thorough yet accessible introduction to the MCES algorithm and known results for it. The theoretical results are also clear and intuitive, as well as their proofs. I agree with the claim that the main advantage here compared to previous work is in using all states of the trajectory for the update, instead of only the first one.

With that said, this seems like a classic case where the simplicity and elegance of the theory is because it is indeed simple and easy to derive. The results are intuitive and not surprising since they are (in my opinion) almost trivial. Once you limit the environment to a DAG and are able to sample each state-action pair indefinitely, it seems obvious to expect to converge to the optimal policy. This follows from the basic dynamic programming concept where you begin from the last state and gradually update your value function to being optimal. Also, as the limited subclass of MDPs considered which are essentially DAGs, it feels like classic CS literature should have similar solutions so a literature survey in that area is advised.

I think the contribution would have been more significant if, for instance, the indefinite initial state selection would have not been possible. In that case, you will not necessarily sample the terminal state or its neighbours and a probabilistic analysis would most likely be needed. Alternatively, if the rate of convergence would have been analyzed, we could understand how the sampling procedure affects convergence, which might help guide more efficient variants of the algorithm.

Lastly, two questions: I'm not sure that AlphaZero updates all states within a trajectory instead of just the initial one as claimed in the middle of p.2. A reference is missing there. Second, why do mujoco environments qualify as a feed-forward environment as claimed in the bottom of p.2? The walking tasks there are highly repetitive.

**Summary Of The Paper:**

The paper studies the Monte-Carlo Exploring-Starts algorithm on MDPs where states are never re-visited and provides convergence results for such MDPs.

**Summary Of The Review:**

To summarize, while I enjoyed reading the paper and its simplicity (which I appreciate in general), its contributions are correspondingly minor and not surprising. The combination of the highly restrictive realm of tabular MDPs in which states are never revisited, together with the strong assumption of ability to sample every state-action indefinitely, generate an easy problem that is not particularly hard or interesting to solve.

---

> ### Author Response · Authors · 2021-11-16
> **Response to Reviewer U5JN**
>
> Thank you for your very detailed comments and questions! Here are our response to your concerns:
>
> > The paper is written well; I enjoyed how it provides a thorough yet accessible introduction to the MCES algorithm and known results for it. The theoretical results are also clear and intuitive, as well as their proofs. I agree with the claim that the main advantage here compared to previous work is in using all states of the trajectory for the update, instead of only the first one.
> >
> > With that said, this seems like a classic case where the simplicity and elegance of the theory is because it is indeed simple and easy to derive. The results are intuitive and not surprising since they are (in my opinion) almost trivial. Once you limit the environment to a DAG and are able to sample each state-action pair indefinitely, it seems obvious to expect to converge to the optimal policy. This follows from the basic dynamic programming concept where you begin from the last state and gradually update your value function to being optimal. Also, as the limited subclass of MDPs considered which are essentially DAGs, it feels like classic CS literature should have similar solutions so a literature survey in that area is advised.
>
>
> Thank you for your comments. We first want to point out that our most general assumption is that the MDP is Optimal Policy Feed Forward (OPFF), for which it is only required that states are never revisited for **optimal policies**. States may be revisited for all other policies. We provide many non-trivial examples of OPFF MDPs in the paper. In fact, we believe most episodic MDPs of practical interest are OPFF.
>
> Second, we do not agree that almost sure convergence is obvious for OPFF MDPs. We don’t think it was even obvious to Sutton and Barto that MCES converges almost surely for Blackjack; otherwise, the authors would have mentioned this in the textbook. Also note that DP methods such as value iteration do not apply to our setting where a perfect model of the environment is not given. Third, although the proof does not require sophisticated tools such as stochastic approximations, the almost-sure convergence proofs are nevertheless non-trivial. We invite the reviewer to read through them carefully.
>
> Also, it is important to keep in mind that the convergence of MCES has been a long-standing open problem, and as discussed in the Introduction, is a nuanced issue since convergence is not guaranteed as it is for Q-learning. It is therefore important to prove convergence for a large and important class of MDPs.
>
> > I think the contribution would have been more significant if, for instance, the indefinite initial state selection would have not been possible. In that case, you will not necessarily sample the terminal state or its neighbours and a probabilistic analysis would most likely be needed. Alternatively, if the rate of convergence would have been analyzed, we could understand how the sampling procedure affects convergence, which might help guide more efficient variants of the algorithm.
>
> Our goal is to study the MCES algorithm, for which you start episodes at every initial state infinitely often. As mentioned in the Introduction, convergence is not guaranteed for MCES. Our goal is not to understand rates of convergence, but instead for what types of MDPs convergence occurs. We have shown convergence will indeed occur for a large and important class of MDPs.

---

> > ### Author Response · Authors · 2021-11-16
> > **Response to Reviewer U5JN (continued)**
> >
> > > Lastly, two questions: I'm not sure that AlphaZero updates all states within a trajectory instead of just the initial one as claimed in the middle of p.2. A reference is missing there. Second, why do mujoco environments qualify as a feed-forward environment as claimed in the bottom of p.2? The walking tasks there are highly repetitive.
> >
> > Thank you for the questions. For AlphaZero, indeed all states within a trajectory are updated (the details of the algorithm are explained on page 2 of [1]). It might be easier to see this when we consider the game of Go: for each episode, the initial state is always an empty Go board with no pieces, and then along the trajectory, the agent transitions into new states. In this case, if only the initial state in each episode is updated, then the agent will only be able to learn about one state, which does not make sense. We will add reference and more discussion on AlphaGo in our revision.
> >
> > For the MuJoCo environment, note if we treat it as an **infinite-horizon** task (for example, if the task is to keep running indefinitely), then MuJoCo is indeed not optimal policy feed-forward (OPFF) since it is not episodic. However, if we treat MuJoCo as an **episodic** MDP (for example, if the task is to run towards a goal, and the episode terminates when the goal is reached), then it falls into the category of OPFF because the simulation is **deterministic**. All deterministic episodic MDPs are OPFF. Thank you for bringing our attention to this detail, we will make this part more clear in our revision.
> >
> > [1] Mastering the game of go without human knowledge, Silver et al.

---

> > > ### Comment · Reviewer_U5JN · 2021-11-28
> > > **Response to rebuttal**
> > >
> > > Thank you for clarifying some of the questions I raised. However, I want to comment that I strongly disagree with the point you're making regarding AlphaZero. In your explanation here, you ignored the fact that MCTS is the basis of the RL procedure. So, if we inspect AlphaZero as an MCES algorithm, then the initial state changes each time according to our position in the tree and thus *it is not* the empty Go board. This initial state is a different one per each iteration of tree expansion.

---

> > > > ### Author Response · Authors · 2021-11-29
> > > > **Updated response to Reviewer U5JN**
> > > >
> > > > Thank you for your comment! The AlphaZero learning process can be roughly seen as a nested loop: there is the outer loop where the agent starts the game from an empty board and plays to the end of the game, and then for each state s on this trajectory, there is the inner loop of MCTS, which is a planning phase that starts from state s. After the planning finishes, a single physical move is made (in the outer loop).
> > > >
> > > > So in our previous response, we are only looking at AlphaZero from a high level, and if we focus on the MCTS process at the inner loop, then you are absolutely correct that the initial state changes and we fully agree with you.
> > > >
> > > > Apparently AlphaZero is very different from MCES (even the MCTS algorithm alone is very different from MCES), so we are not saying they do the same thing, but only that they share some important similarities on a very high level. We hope that clarifies things and please let us know if you have further questions!

---

> ### Author Response · Authors · 2021-11-24
> **Additional response to Reviewer U5JN**
>
> Thank you for your comments again! We would like to post another response to further address your concern on the significance of our proof. In our paper, the proof for the OPFF MDPs can be found in the appendix; however, we only put it in the appendix because we did not have enough space in the main paper.
>
> We want to emphasize that the proof for OPFF MDPs is very important, and it is also  harder and more nuanced than the proof for the simpler SFF MDPs, which is itself nuanced and challenging, although only making use of the strong law of large numbers.
>
> At a very high level, in the SFF proof (in the induction step) we show that for a state s, if all actions from this state lead to states for which the policy and Q values have nearly converged, then we can use a variant of SLLN to show that the Q value for all actions in state s will converge almost surely.
>
> Note that in the OPFF MDPs, a non-optimal action can lead to an arbitrary state in the state space. As a consequence, for a non-optimal action, if the policy has not converged for **all states**, then a non-optimal action can lead to returns with **arbitrary values** (note these returns will not have a fixed mean). This immediately makes it impossible to apply the same proof, since now for a state s, we simply cannot directly establish the convergence of Q value for all actions. This further makes it difficult to show that the policy indeed converges almost surely.
>
> So the OPFF proof has to be done in a more careful and sophisticated way. On a very high level, we need to perform a 2-stage induction: in the first stage, we show that the Q estimate for the optimal action in each state will almost surely converge to the optimal Q value, and the policy for each state will converge to the optimal policy; in the second stage, we show that the Q values for all actions (including the non-optimal actions) converge almost surely to their optimal Q value. We also need to do the proof very carefully because the MDP is stochastic. (Note that even if the MDP is deterministic, such a 2-stage proof is still necessary, but will be simpler compared to the stochastic case)
>
> We hope this response will help clarify the significance of our theoretical results. We appreciate your time and we are happy to discuss with you if you have any further questions.

---

### Official Review · Reviewer_Fsd8 · 2021-10-31

**Correctness:** 4
**Technical Novelty And Significance:** 4
**Empirical Novelty And Significance:** 3
**Recommendation:** 8
**Confidence:** 4

**Main Review:**

Main comments:

Overall I really like this paper. To prove convergence of stochastic iterative algorithms, existing results rely on the properties of Bellman equation, and use either super-martingale convergence or ODE approach. This paper uses induction along with strong law of large numbers, resulting in a much simpler proof. This paper does seem to be of interest to the broader community of ICLR.

Other comments/questions:

1 There are several typos, such as (1) "Corrolary 1" should be replaced by "Corollary 1", (2) the sentence before Corollary 1, the "snd" should be replaced by "and". The authors should carefully go over the paper again to fix them.

2 As for the numerical experiments, I am curious to see the convergence rate comparison between MCES and model-free Q-learning, which is probably the most popular value-based RL algorithm in the literature.

Suggestions on future directions: Now that convergence is shown for tabular MCES algorithm, there are many interesting potential future directions for this line of research.

1 Convergence rate. The convergence rate of strong law of large numbers has been established in the literature. While the induction technique can be used to show asymptotic convergence, it is not clear if it can be used to show the convergence rate. It is an interesting future direction to show the convergence rate of MCES and compare it to that of Q-learning.

2 Function approximation. The next question is about extending the result to the function approximation setting. Q-learning with function approximation is an major theoretical open problem in RL because of the deadly triad. I am curious to see if one can show any theoretical guarantees on MCES with function approximation, or if it suffers from the same difficulty as Q-learning.

-------------After Author Feedback-------------

Thank the authors for their feedback. I would like to keep my score and vote for acceptance.




**Summary Of The Paper:**

This paper studies Monte Carlo with exploration starts algorithm for solving the reinforcement learning problem. The writing is clear and I enjoyed reading this paper. As for the results, asymptotic convergence of the algorithm is established without needing strong assumptions in related literature. As pointed out by the authors, the result resolves an important open problem in RL. The proof is simple and intuitive. Numerical experiments corroborate theoretical findings.

**Summary Of The Review:**

This paper provides convergence guarantees of MCES algorithm under mild assumptions, hence resolving an important open problem in RL. I enjoy reading this paper and I think it has enough contribution to be published at ICLR.

---

> ### Author Response · Authors · 2021-11-16
> **Response to Reviewer Fsd8**
>
> Thank you for your positive evaluation and insightful suggestions! Here are our response to your comments:
>
> > 1 There are several typos, such as (1) "Corrolary 1" should be replaced by "Corollary 1", (2) the sentence before Corollary 1, the "snd" should be replaced by "and". The authors should carefully go over the paper again to fix them.
>
> Thank you for pointing out the typos. The typos will be fixed in the revision.
>
> > 2 As for the numerical experiments, I am curious to see the convergence rate comparison between MCES and model-free Q-learning, which is probably the most popular value-based RL algorithm in the literature.
>
> Thanks for the suggestion. We will run additional experiments and present the results in the appendix.
>
> > Suggestions on future directions: Now that convergence is shown for tabular MCES algorithm, there are many interesting potential future directions for this line of research.
> >
> > 1 Convergence rate. The convergence rate of strong law of large numbers has been established in the literature. While the induction technique can be used to show asymptotic convergence, it is not clear if it can be used to show the convergence rate. It is an interesting future direction to show the convergence rate of MCES and compare it to that of Q-learning.
> >
> > 2 Function approximation. The next question is about extending the result to the function approximation setting. Q-learning with function approximation is an major theoretical open problem in RL because of the deadly triad. I am curious to see if one can show any theoretical guarantees on MCES with function approximation, or if it suffers from the same difficulty as Q-learning.
>
> We agree with you that these are interesting and important research directions. We will be looking into these and related research directions in the near future.

---

### Decision · Program_Chairs · 2022-01-20

**Decision:**

Accept (Poster)

**Comment:**

The paper considers the convergence of the Monte Carlo Exploring Start (MCES) algorithm, a basic method in RL. Although the method is very simple and known for a long time, the condition for its convergence is not completely understood.

One of the latest results is from almost 20 years ago (Tsitsiklis, "On the Convergence of Optimistic Policy Iteration," 2002). That paper shows the convergence of the method under some restrictive assumptions on how the algorithm operates. In particular, that result requires that only the action-value function of the initial state-action pair of an episode be updated, as opposed to all visited state-action pairs. What is not known is whether we can update the action-values of all state-action pairs observed in a trajectory.

It is notable that one of Tsitsiklis (2002) result requires a uniform selection of the initial state-action. But he also describes a modification of the algorithm that allows convergence with a non-uniform initial distribution (see page 66 of that paper, close to the end of Section 3 - Optimistic Policy Iteration Using Monte Carlo for Policy Evaluation).

On the other hand, this paper establishes the convergence of MCES with no assumption on how the algorithm works. In particular, the algorithm updates the action-value function for all state-action pairs observed in an episode. As long as all state-action pairs are visited infinitely often (and not necessarily from a uniform distribution), the convergence is established.

There is a catch, however. The paper requires some assumptions on the class of MDPs. In particular, it requires the environment to be either Stochastic Feed-Forward (SFF) (more restrictive) or Optimal Policy Feed-Forward (OPFF) (less restrictive). The OPFF assumption states that under any optimal policy, a state is never revisited within an episode.

The proof technique of this paper is different from the usual stochastic approximation method, and may be considered simpler.

We have two positive reviewers (with score of 8) and two slightly negative ones (with score of 5). The concern of negative reviewers is that the OPFF assumption is very restrictive. And given such an assumption on the class of MDPs, the proof becomes very simple. Moreover, it is not clear that the proof techniques developed in this work is a step toward analyzing MCES for more general MDPs.

My related concern is whether OPFF vs. Non-OPFF is a good way to characterize MDPs for which MCES with every-state update is convergent. Figure 3 in the paper shows the current state of knowledge in the analysis of variants of MCES. We know a problem with "No convergence" within the fourth quadrant (which is Non-OPFF).
Is the class of Non-OPFF a tight superset of the non-convergent ones?  Or is it a much larger superset? If it is tight, then OPFF is a good characterization of when MCES works or not. If it is not tight, OPFF may not be the right way to characterize the convergence of MCES.

All being said, I believe this paper positively contributes to our knowledge of a basic and fundamental RL algorithm. It does not fully resolve the convergence question, but it is indeed a progress. Whether OPFF characterization is going to be the right one or not remains to be seen in the future. I would act optimistically here and recommend acceptance of the paper.

I encourage the authors to incorporate the comments by the reviewers in updating their papers, including:
- fixing all the typos
- providing more examples of problems that are OPFF
- resolving the claim about alphaZero
- providing empirical comparison with Q-Learning